# A Systematic Approach to Identify and Manage Interface Risks between Project Stakeholders in Construction Projects

Michael C. Okika [1,*], Andre Vermeulen [1] and Jan-Harm C. Pretorius [2]

1   Postgraduate School of Engineering Management, University of Johannesburg, Auckland Park, P.O. Box 524, Johannesburg 2006, South Africa; avermeulen.research@gmail.com
2   Faculty of Engineering and the Built Environment, University of Johannesburg, Auckland Park, P.O. Box 524, Johannesburg 2006, South Africa; jhcpretorius@uj.ac.za
*   Correspondence: michaelieece@yahoo.co.uk or 219123868@student.uj.ac.za

**Abstract:** Interface risks are inherent in every construction project from start to finish. Identifying and managing these risks effectively in every project phase is crucial for actualising project objectives. This paper shows a comprehensive framework showing several relationships between project stakeholders and how the interface risks between them that influence project execution are identified and managed for the overall construction project success. Firstly, a literature review on interfaces and interface risks and a discussion on how organisations managed interface risks were carried out, and secondly, the collection of quantitative data was conducted by means of structured online questionnaires. The sample consisted of 205 construction project professionals who were selected randomly. This group included individuals with various roles in the construction industry. The data were analysed using descriptive statistical methods, including factor analysis, reliability assessment, and calculations of frequencies and percentages. The results showed all the factors, work cultures, and organisational approaches that influence interface risk management and ways to identify and manage interface risks effectively. Effective stakeholder management is crucial for effective interface risk management since many interface risks are created by the numerous stakeholders involved in the project and the proposed frameworks will effectively mitigate the consequences and causes of interface risks. Effectively mitigating these risks involves effective stakeholder management, building information modelling volume strategy, and creating a virtual construction model during the construction phase; in addition, construction supply chain risks must be carefully identified during the interfaces establishment stages; interface risks must be carefully identified during the conceptualisation; and the planning, construction, and execution stages and standard methods and procedures must be defined to effectively identify and manage interface risks as the occur in the project lifecycle plus implementing the proposed risk mitigation frameworks.

**Keywords:** interface risk management; project stakeholders; construction; risk; interfaces; interface risks





## 1. Introduction

In the present economic age, a sizable proportion of huge construction projects has been steadily expanding. Concurrently, the pandemic-induced economic crisis in 2020 strongly demonstrated that successful project delivery is dependent on sustainable project lifecycle management as well as project management principles and methodologies [1]. Interfaces are points of interaction between two or more aspects of a project which might be between clients, contractors, subcontractors, and other project stakeholders. Poor interface management gives rise to interface risks. Interface risks and the failure to manage them effectively is a common cause of problems in construction projects, which can negatively affect project objectives and goals.

The construction industry encounters interface risks which are complex, difficult, and require diverse solution to solve and manage, involving interface risk management

(IRM). Interface risks are the most commonly encountered problem in the industry. In the highly risky and complex environment of a construction project, if effective decisions are not made in the conceptualisation, planning, design, contracting, procurement, and execution phases, then disagreements, loss of profit, claims, industrial actions, disputes, conflicts, change orders, and claims can occur at any phase of the construction project. The traditional construction industry usually depends on the project participants' work experiences to solve interface risk problems, including issues between designers, owners, project team members, main contractors, subcontractors, host communities, licensing and regulatory bodies, vendors, maintenance contractors, and material suppliers. Formal risk management plans are rarely used in executing construction projects because contractors and subcontractors rely on their past experiences and judgements, and this can result in unforeseen circumstances that can negatively impact project objectives since these individuals are not fully equipped with the tools to effectively manage the unidentified risks and uncertainties associated with interfaces and numerous stakeholders. Stakeholder management is often not effectively incorporated into risk management plans as the numerous stakeholders and their roles and influences are not carefully identified in the conceptualisation stage of a project, and this gives rise to additional interface risks.

Interface risk management is primarily overseen and regulated by project managers. However, the intricate handling of these interface incidents is frequently evaluated and appraised based on the expertise of engineers. The involvement of a systematic approach to interface problems is infrequent. In essence, the conventional approach to interface problem solving lacks objectivity, relies heavily on subjective experiences, and lacks a systematic framework for identifying interface issues and proposing comprehensive solutions. The professionalisation of interface risk management (IRM) practise has been shown to have a positive impact on the project performance of construction projects [2]. This, in turn, leads to enhanced social benefits for public projects. While the advantages of interface risk management (IRM) may be more readily apparent in large-scale projects, the effective management of interfaces is considered significant for projects of all sizes and levels of complexity. Furthermore, recent research conducted by [3,4] has revealed that project managers have utilised building information modelling (BIM) to effectively oversee extensive construction projects and address the challenges associated with interfaces. In addition to its academic significance, this study also demonstrates its social relevance by potentially contributing to the professionalisation of IRM. The academic literature suggests that IRM holds promising benefits. One can anticipate several benefits from improving the exchange of information and reducing costs associated with interface issues, such as the promotion of inter-organizational collaboration [2,5].

Construction projects employ principles and protocols that encompass a multitude of complexities in the management of various stakeholders, including owners; technical clients; and engineering, procurement, and construction (EPC) contractors. The reason for this is that the phases of the construction project encompass numerous contracts that involve a diverse range of contractors. According to [6], therefore, it is important to recognise that the application of principles and approaches may vary among different stakeholders, both internal and external. Firstly, it is not feasible to effectively manage the relational connections between a singular project team consisting of the general contractor, client, designer, and customer [7]. Furthermore, the premise of a singular project team is predicated on the explicit consideration of the individual interests and objectives of all the participants [8]. In practical applications, the interests of the individuals engaged in a construction endeavour exhibit variation and frequently encompass multiple facets. This phenomenon arises in scenarios where the proprietor aims to reduce the expenses associated with construction, while the general contractor or subcontractor seeks to augment the construction costs. Additionally, the technical customer plans to delegate the tasks and coordination work to the design firm, thereby necessitating supplementary compensation [9]. According to [10], when considering the selection of the most economically efficient alternatives, the practicality of implementing a sole project team is questionable. The contractor expresses a

favourable perspective regarding the evaluation of the most financially advantageous construction project. Nevertheless, the limited availability of construction orders to contractors can be attributed to competition from other industry players and market conditions. The primary concern for customers is the fulfilment of technical construction orders. Interface risk management is commonly employed in intricate projects and overseen by multiple stakeholders with diverse areas of expertise, resulting in a multitude of overlapping activities. Stakeholder management is one of the major challenges faced in construction projects as these numerous stakeholders have personal and diverse interests and objectives ranging from personal monetary gains, political and economic interests, opportunities, and favours and these, in essence, compromise the objectives of the project to successfully and in a timely fashion complete the project. This paper identifies and proposes new methods to manage stakeholders and, in essence, mitigate interface risks from these stakeholders and interfaces.

Interface risk management is a potential solution for effectively managing the complexities of construction projects. It primarily involves the management of communications, relationships, and deliverables among project stakeholders. By establishing improved methods for identifying, documenting, monitoring, and tracking project interfaces and the associated risks, interface risk management can contribute to the successful execution of construction projects. The present study undertakes a comprehensive review of relevant literature in order to establish a solid theoretical foundation for the research. The term "interfaces" in the context of construction projects refers to the points of connection or interaction between different components, systems, or stakeholders involved in the project. These interfaces play a crucial role in ensuring the successful coordination and integration of various elements within the construction process. Interfaces are significant for the overall project execution.

The study objective was to carry out a literature review on interfaces in construction, as well as interface risks and interface risk management. The study three main objectives were:

1. To identify the consequences of poor and ineffective interface risk management approaches and how they influence construction project delivery.
2. To identify the current interface risk management methods utilised by organisations.
3. To identify the causes of interface risks and how they influence project objectives.

To support the objectives of the study, these three research questions were asked. Respondents were asked to identify:

1. The causes of interface risks.
2. The consequences of poor and ineffective interface risk management approaches.
3. The interface risk management approaches implemented by their organisations.

The study focuses on a systematic approach of identifying and managing risks associated with every interface in construction projects in every phase. The literature review was carried out to identify critical areas of knowledge in the field of study, with the purpose of presenting a summary of the recent literature on the topic. The primary objective of the study was to develop a framework on how to identify and manage interface risks in construction for overall project success.

This paper is composed of the following: the study background is discussed after the Introduction, then Section 3 discusses the materials and methods used, Section 4 encompasses the findings and analysis, and the results and discussion are given in Section 5, followed by the conclusion in Section 6.

## 2. Background

### 2.1. Definition and Significance of Interfaces in Construction Projects

The concept of the interface was initially introduced by Wren, D.A., within the realm of organisational management. It was defined as the point of contact between interacting organisations that possesses a certain degree of autonomy. According to [11], there is a need to prioritise factors such as information sharing, degree of cooperation, and

response time among organisational interfaces in construction projects. The concept of interface management encompasses the effective information management, coordination, and responsibility across contractual, physical, and organisational boundaries. It is widely recognised as a valuable approach for fostering friendly collaboration between project organisations within the construction industry [12]. The effective management of interfaces in the construction industry is widely recognised as a socially oriented activity that extends beyond formal practises and procedures [5]. In the context of interface classification, [13] employed the term "internal" to denote interactions occurring exclusively within the confines of a single project environment. Conversely, the term "external" refers to relationships established with entities that have no direct involvement in the project. In a survey conducted by [13], a range of interface issues were identified by industry experts. These issues included permits, change orders, contract obligations, poor quality of works, government laws, environmental problems, long lead items, poor contracting strategy, and wrong specifications.

## 2.2. Interface Risk Management

According to [13], there exists a differentiation between interface management and integration management. Integration management primarily concerns itself with the coordination of various project elements, encompassing the associated processes. On the other hand, interface management primarily involves the identification of stakeholder points of contact and the associated risks. According to scholars in the construction industry, interface management is widely recognised as a means to enhance goal alignment, mitigate conflicts, and improve cooperation efficiency among participants. Considering the evident significance of systems thinking in addressing interfaces, it was anticipated that the existing body of general systems engineering (SE) literature would offer comprehensive information on the organisation of information management (IM). Contrary to the previous statement, the opposite holds true. The book authored by Hsu (2020) regarding the foundations of software engineering in industrial practise exhibits limited focus on the subject matter. The primary emphasis of this study pertains exclusively to physical interfaces, encompassing their identification using various tools and their management through control documents. Hence, it is understandable that scholars advocate for the formalisation of interface management through the implementation of a methodical approach. As a result, recent scholarly endeavours have predominantly concentrated on the advancement of formal governance approaches through the utilisation of standardised procedures and information technology [2,5]. According to [5], research indicates that individuals involved in projects lack a comprehensive understanding of the necessary components for proficiently managing interfaces. People prefer to collaborate with people they like and trust, and they are more tolerant of conflict and differences. Controlling and supervisory behaviours emerge when people cooperate with someone they do not trust. This circumstance, known as Guanxi, is common in Chinese culture and has the potential to foster widespread social relationships [14]. Guanxi, as hypothesized by [15], can attenuate the influence of contractual control on project conflict, demonstrating that human interactions have a significant impact on inter-organizational trust relationships. Owners in the construction business frequently expect that contractors will engage in opportunistic conduct at any cost. Contractors, on the other hand, strive to delight owners in order to build trust in their organizations [16].

The implementation of practical guidelines has the potential to have a positive impact on individuals' behaviours towards interface management. Additionally, it can foster a collective comprehension of interface management, which is considered crucial for enhancing its application [5]. According to [2], there is a positive correlation between the enhanced construction project outcome and the improved interface risk management performance.

Coordination-related issues, such as improper communication, the mismatch of owner expectations, rampant bureaucracy, dishonest practices, and disputes, are frequently observed on projects. Such coordination challenges necessitate synergies through continuous interface management [17–20]. Trust, openness, and communication are believed to aid

in achieving strong interface management performance in terms of quality, time, and cost. The foundation of interface management is trust between parties. Project participants are more likely to be open to share their valuable resources if they have trust and mutual commitment, allowing for more porous and flexible organizational boundaries. However, simply being open is not enough to obtain high interface management performance. To communicate correct and comprehensive interface information and to promote efficient information sharing, actions such as building effective communication channels that adapt to organizational structure and characteristics are required. It is critical to design moderate interface mechanisms for interdisciplinary and multiorganizational communication amongst parities in order to achieve timely communication, coordination, and cooperation [21]. Transaction cost is one of the contributing factors to interface risks, and according to the Standish Group's released reports, project costs are routinely surpassed by 21% to 100% (60% of project cost overruns). Furthermore, 4% of initiatives had a cost overrun that cause the project's final cost to be higher than four times previously predicted. Standish believed that the group analysis of construction projects in the United Kingdom revealed cost variances ranging from 50 to 80%, which is significant for a construction project [22,23]. According to [23], some of the factors that affect transaction costs are payment schedule, quality of project, type of project, value of project, complexity of the project, similar experience, parties' relationships, change order, the efficiency of the organisation, the duration of the project, and many more, and if these factors are not carefully considered during the project lifecycle, the projects will be adversely affected. Some of these costs are incurred to secure the delivery of high-quality developments or environmental enhancement, but it is often evident that the costs are exorbitant and the procedure is overly long and inefficient. Identifying and decreasing such costs assists planners and decision makers in improving the efficiency, efficacy, and acceptability of their projects and processes [24–26].

## 3. Materials and Methods

The primary data were collected from project managers, civil/structural engineers, mechanical engineers, risk managers, architects, quantity surveyors, electrical engineers, construction managers, HSE managers, estate managers, and other construction industry professionals actively working in construction projects in South Africa through an online questionnaire developed specifically for this study in order to answer the research questions and to realise the research objectives. Secondary data were collected through a review of the relevant literature, articles, and journals in the construction industry. A total of 205 research questionnaires were distributed to participants active in the construction industry. The study focused only on the South African construction industry and active industry professionals. All respondents were South African residents and had experience in the South African construction industry. Industry professionals living outside of South Africa were not part of the study. Every participant was involved in the conceptualisation, planning, contracting, subcontracting, procurement, construction, execution, HSE management, or commissioning phases of a project. These three Likert-type scale response anchors were chosen for the questionnaire in order to find out the level of agreement with the individual statements in the questionnaire and the frequencies of each statement or items in the questionnaire, and the extent scale was used to find out the extent to which each statement or item in the questionnaire influences construction projects. The data collection process commenced by administering a biographical questionnaire to ascertain the appropriate research participants in Section A; this included size of the organisation, profession, age, and highest academic qualifications. Section B had three subsections, namely B2, B3, and B4. Section B2 involved questions related to the consequences of poor and ineffective interface risk management approach, where respondents were asked to identify these consequences and rate them according to the extent scale. Section B3 comprised questions related to interface risk management methods currently adopted by their organisations and the extent to which they influenced project objectives, and Section B4 comprised questions related

to the causes of interface risks and the respondents were asked to identify the extents the causes of interface risks influenced the successful execution of construction projects.

The items or questions in each section were coded for ease of analysis. For the five-point Linkert scale chosen, "rarely" was coded as a 2, "sometimes" was coded as a 3, "often" was coded as a 4, and "always" was coded as a 5. Also, "strongly disagree" was coded as a 1, "disagree" was coded as a 2, "neutral" was coded as a 3, "agree" was coded as a 4, and "strongly agree" was coded as a 5. In addition, "to no extent" was coded as a 1, "to a small extent" was coded as a 2, "to a moderate extent" was coded as a 3, "to a large extent" was coded as a 4, and "to a very large extent" was coded as a 5.

The data obtained from the questionnaire were coded, recorded, and analysed utilising the Statistical Package for the Social Sciences (SPSS, V25). Factor analysis was conducted in order to identify the latent dimensions underlying the measured variables, as these variables are expected to exhibit correlations or anticipated correlations. This study aims to assess the impact of measured variables and examine the interrelationships among a predetermined set of defined, observed, and quantifiable constructs. According to the guidelines provided in the SPSS manual, the Kaiser–Meyer–Olkin (KMO) measurement and the Bartlett's test of sphericity were employed to assess the suitability of the correlation matrix as an identity matrix, thereby determining the appropriateness of the factor model.

## 4. Findings and Analysis

The study employed the Kaiser–Meyer–Olkin (KMO) measurement and Bartlett's test to assess the interrelationships among variables, thereby informing the decision to proceed with the factor analysis of the collected data. A comprehensive set of 205 responses was obtained from the designated target population, which primarily comprised individuals within the construction industry, as described in the context of questionnaire design and target group identification. Table 1 below shows the summary of the biographical data of the respondents who participated in the online survey.

**Table 1.** Survey participants' professions in the South African construction industry.

| Profession | Frequency | Percent | Valid Percent | Cumulative Percent |
|---|---|---|---|---|
| Quantity surveyor | 16 | 7.8 | 7.8 | 7.8 |
| Architect | 9 | 4.4 | 4.4 | 12.2 |
| Civil engineer/structural engineer | 27 | 13.2 | 13.2 | 25.4 |
| Builder | 7 | 3.4 | 3.4 | 28.8 |
| Construction manager | 25 | 12.2 | 12.2 | 41.0 |
| Electrical engineer | 22 | 10.7 | 10.7 | 51.7 |
| Mechanical engineer | 20 | 9.8 | 9.8 | 61.5 |
| Estate manager | 8 | 3.9 | 3.9 | 65.4 |
| Project manager | 18 | 8.8 | 8.8 | 74.1 |
| Construction engineer | 13 | 6.3 | 6.3 | 80.5 |
| Project engineer | 8 | 3.9 | 3.9 | 84.4 |
| Project administrator | 9 | 4.4 | 4.4 | 88.8 |
| Safety officer/engineer/manager | 10 | 4.9 | 4.9 | 93.7 |
| Risk manger | 10 | 4.9 | 4.9 | 98.5 |
| Other construction professionals | 3 | 1.5 | 1.5 | 100.0 |
| Total | 205 | 100.0 | 100.0 | |

As seen in Table 1 above, out of the 205 responses from the online questionnaire, 16 respondents were quantity surveyors, 9 were architects, 7 were builders, 8 were project engineers, 9 were project administrators, 10 were safety officers/engineers/managers, 10 were risk managers, 20 were mechanical engineers, 13 were construction engineers, 18 were project managers, 8 were estate managers, 22 were electrical engineers, 25 were construction managers, 27 were civil/structural engineers, and 3 respondents were other construction professionals. Table 2 below shows age distribution of participants.

**Table 2.** Age distribution of respondents.

| Age Group | Frequency | Percent | Valid Percent | Cumulative Percent |
|---|---|---|---|---|
| 21–25 years | 4 | 2.0 | 2.0 | 2.0 |
| 26–30 years | 16 | 7.8 | 7.8 | 9.8 |
| 31–35 years | 32 | 15.6 | 15.6 | 25.4 |
| 36–40 years | 42 | 20.5 | 20.5 | 45.9 |
| 41–45 years | 53 | 25.9 | 25.9 | 71.7 |
| 46 years and above | 58 | 28.3 | 28.3 | 100.0 |
| Total | 205 | 100.0 | 100.0 | |

As seen in Table 2 above, out of the 205 respondents, 4 respondents were in the age group of 21–25 years, 16 were in the age group of 26–30 years, 32 were in the age group of 31–35 years, 42 were in the age group of 36–40 years, 53 were in the age group of 41–45 years, and 46 respondents were in the age group of 46 years and above. Table 3 below shows the academic qualifications of the respondents.

**Table 3.** Academic qualifications of the respondents.

| Highest Academic Qualification | Frequency | Percent | Valid Percent | Cumulative Percent |
|---|---|---|---|---|
| Post-Matric Certificate or Diploma | 11 | 5.4 | 5.4 | 5.4 |
| Bachelor's degree | 55 | 26.8 | 26.8 | 32.2 |
| Honours Degree | 28 | 13.7 | 13.7 | 45.9 |
| Master's degree | 70 | 34.1 | 34.1 | 80.0 |
| Doctorate Degree | 41 | 20.0 | 20.0 | 100.0 |
| Total | 205 | 100.0 | 100.0 | |

As seen in Table 3 above, 11 respondents out of the 205 respondents, which represented 5.4% of the respondents, had post-matric certificates or diplomas as their highest academic qualifications; 55 (26.8%) had Bachelor's degrees; 28 (13.7%) had Honours degrees; 70 (34.1%) had Master's degrees; and 41 respondents, which represented 20.0% of the total respondents, had doctoral degrees. Table 4 below shows the size of the respondents' organizations.

**Table 4.** Size of respondents' organizations.

| Organisation Size | Frequency | Percent | Valid Percent | Cumulative Percent |
|---|---|---|---|---|
| Small (1–100 staff) | 72 | 35.1 | 35.1 | 35.1 |
| Medium (101–500) | 74 | 36.1 | 36.1 | 71.2 |
| Large (501–5000+) | 59 | 28.8 | 28.8 | 100.0 |
| Total | 205 | 100.0 | 100.0 | |

As seen in Table 4 above, 72 respondents, which represents 35.1% of the total respondents, work in the small-sized industries, and 74, which represents 36.1%, work at medium-sized industries, while 59 of the respondents, which represents 28.8%, work in large-scale construction industries.

Table 5 below represents the frequency distribution for question 1 (How often do you encounter interface risks between project stakeholders in a project?)

**Table 5.** Frequency distribution for research question 1.

| How Often Do You Encounter Interface Risks between Project Stakeholders in a Project? | | | | |
|---|---|---|---|---|
| | Frequency | Percent | Valid Percent | Cumulative Percent |
| Rarely | 11 | 5.4 | 5.4 | 5.4 |
| Sometimes | 63 | 30.7 | 30.7 | 36.1 |
| Often | 66 | 32.2 | 32.2 | 68.3 |
| Always | 65 | 31.7 | 31.7 | 100.0 |
| Total | 205 | 100.0 | 100.0 | |

As seen in Table 5 above, 11 (5.4%) respondents chose "rarely", 63 (30.7%) chose "sometimes", 66 (32.2%) chose "often", and 65 (31.2%) of the total respondents chose "always". Table 6 below shows the mean and standard deviation for research question 1.

**Table 6.** Statistics for research question 1.

| How Often Do You Encounter Interface Risks between Project Stakeholders in a Project? | | | | | | | |
|---|---|---|---|---|---|---|---|
| N | | | | | | | |
| Valid | Missing | Mean | Median | Mode | Std. Deviation | Minimum | Maximum |
| 205 | 0 | 3.90 | 4.00 | 4 | 0.913 | 2 | 5 |

As seen in Table 6 above, the mean was 3.90, which was slightly below "often" (4), and most people answered "sometimes" (3) and "always" (5). The median was 4.00, which means that half of the respondents chose between "often" and "always", and the other half chose between "often" and "always". The mode was 4, which means most people chose "often". Table 7 below shows the responses for the research question 2 on work cultures related to interface risks. Interfaces are points of interaction between two or more aspects of a project, which might be between clients, contractors, subcontractors, or other project stakeholders, while interface risks are risks generated because of poor interface management in construction projects.

**Table 7.** Responses on work culture related to interface risks.

| Work Culture Related to Interface Risks | | Strongly Disagree | Disagree | Neutral | Agree | Strongly Agree | Total |
|---|---|---|---|---|---|---|---|
| Interface risks between project stakeholders can be classified as uncertainties. | Count | 1 | 15 | 39 | 129 | 21 | 205 |
| | Row N% | 0.5% | 7.3% | 19.0% | 62.9% | 10.2% | 100.0% |
| Interface risks between project stakeholders can be classified as unidentified risks. | Count | 1 | 12 | 57 | 118 | 17 | 205 |
| | Row N% | 0.5% | 5.9% | 27.8% | 57.6% | 8.3% | 100.0% |
| Identification of hard interface risks encourages effective collaboration between project | Count | 1 | 2 | 11 | 119 | 72 | 205 |
| | Row N% | 0.5% | 1.0% | 5.4% | 58.0% | 35.1% | 100.0% |
| Identification of soft interface risks encourages effective collaboration between project stakeholders. | Count | 0 | 6 | 27 | 107 | 65 | 205 |
| | Row N% | 0.0% | 2.9% | 13.2% | 52.2% | 31.7% | 100.0% |

Table 7 above shows the responses for questions on work culture related to interface risks. The respondents were asked to answer the questions and rank them according to their level of agreement.

For the first statement ("Interface risks between project stakeholders can be classified as uncertainties."), 1 respondent strongly disagreed with the statement, which represents 0.5% of the total responses; 15 (7.3%) respondents disagreed; 39 (19.0%) respondents were neutral; 129 (62.9%) agreed; while 21 (10.2%) of the respondents strongly agreed.

For the second statement ("Interface risks between project stakeholders can be classified as unidentified risks."), 1 respondent strongly disagreed with the statement, which represents 0.5% of the responses; 12 (5.9%) disagreed with the statement; 57 (27.8%) respondents were neutral; 118 (57.6%) agreed; while 17 (8.3%) respondents strongly agreed.

For the third statement ("Identification of hard interface risks encourages effective collaboration between project stakeholders."), 1 respondent strongly disagreed with the statement, which represents 0.5% of the responses; 2 (1.0%) disagreed with the statement; 11 (5.4%) respondents were neutral; 119 (58.0%) agreed; while 72 (35.1%) respondents strongly agreed.

For the fourth statement ("Identification of soft interface risks encourages effective collaboration between project stakeholders."), no respondent strongly disagreed with the statement, which represented 0.0% of the responses; 6 (2.9%) disagreed with the statement; 27 (13.2%) respondents were neutral; 107 (52.2%) agreed; while 65 (31.7%) respondents strongly agreed.

Table 8 below shows the KMO and Bartlett's test for research objective 1 (consequences of poor and ineffective interface risk management approach). KMO seeks to determine the applicability of a result to a set of measures when conducting factor analysis, where the values must be greater than 0.6, while Bartlett's test of sphericity must be less than 0.05 to establish the applicability of factor analysis.

**Table 8.** KMO and Bartlett's test for research objective 1 for B2 (consequences of poor and ineffective interface risk management approach).

| KMO and Bartlett's Test | | |
|---|---|---|
| Kaiser–Meyer–Olkin Measure of Sampling Adequacy. | | 0.898 |
| Bartlett's Test of Sphericity | Approx. Chi-Square | 1309.488 |
| | Df | 78 |
| | Sig. | <0.001 |

As shown in Table 8 above, the Kaiser–Meyer–Olkin measure of sampling adequacy was 0.898, which was greater than 0.6 and demonstrates that factor analysis can be carried out. For the Bartlett's test of sphericity, the significance, which is the $p$ value, was less than 0.001, and thus was less than 0.05, and this supports its factorability. Table 9 below shows the KMO and Bartlett's test for research objective 2 (interface risk management approaches by organisations).

**Table 9.** KMO and Bartlett's test for research objective 2 for B3 (interface risk management approaches by organisations).

| KMO and Bartlett's Test | | |
|---|---|---|
| Kaiser–Meyer–Olkin Measure of Sampling Adequacy. | | 0.915 |
| Bartlett's Test of Sphericity | Approx. Chi-Square | 4068.497 |
| | Df | 276 |
| | Sig. | 0.000 |

As shown in Table 9 above, the Kaiser–Meyer–Olkin measure of sampling adequacy was 0.915, which was greater than 0,6, and therefore, factor analysis can be carried out. For Bartlett's test of sphericity, the significance, which is the $p$ value, is 0.000, which is less than 0.05, and this supports its factorability. Table 10 below represents KMO and Bartlett's test for research objective 3 (causes of interface risks).

**Table 10.** KMO and Bartlett's test for research objective 3 for B4 (causes of interface risks).

| KMO and Bartlett's Test | | |
|---|---|---|
| Kaiser–Meyer–Olkin Measure of Sampling Adequacy. | | 0.917 |
| Bartlett's Test of Sphericity | Approx. Chi-Square | 2767.160 |
| | Df | 171 |
| | Sig. | 0.000 |

As shown in Table 10 above, the Kaiser–Meyer–Olkin measure of sampling adequacy was 0.917, which was greater than 0.6 and shows that the factor analysis can be carried out. For Bartlett's test of sphericity, the significance, which is the $p$ value, was 0.000, which is less than 0.05, and this supports its factorability. Table 11 below shows the responses received for research objective 1 for B2, which is related to consequences of a poor and ineffective interface risk management approach. The respondents were asked to rank these consequences according to the extent scale.

**Table 11.** Responses for the research objective 1 for B2—the consequences of a poor and ineffective interface risk management approach.

| Section B2 (B2.1–B2.13) | | To no Extent | To a Small Extent | To a Moderate Extent | To a Large Extent | To a Very Large Extent | Total |
|---|---|---|---|---|---|---|---|
| Stakeholders' complaints | Count | 1 | 5 | 42 | 101 | 56 | 205 |
| | Row N% | 0.5% | 2.4% | 20.5% | 49.3% | 27.3% | 100.0% |
| Claims for damage | Count | 0 | 8 | 40 | 94 | 63 | 205 |
| | Row N% | 0.0% | 3.9% | 19.5% | 45.9% | 30.7% | 100.0% |
| Loss of profit | Count | 1 | 2 | 18 | 87 | 97 | 205 |
| | Row N% | 0.5% | 1.0% | 8.8% | 42.4% | 47.3% | 100.0% |
| Reputational damage of an organisation | Count | 0 | 8 | 40 | 89 | 68 | 205 |
| | Row N% | 0.0% | 3.9% | 19.5% | 43.4% | 33.2% | 100.0% |
| Industrial actions | Count | 0 | 12 | 60 | 88 | 45 | 205 |
| | Row N% | 0.0% | 5.9% | 29.3% | 42.9% | 22.0% | 100.0% |
| Project delays | Count | 2 | 2 | 22 | 102 | 77 | 205 |
| | Row N% | 1.0% | 1.0% | 10.7% | 49.8% | 37.6% | 100.0% |
| Regulatory infringements | Count | 2 | 4 | 77 | 86 | 36 | 205 |
| | Row N% | 1.0% | 2.0% | 37.6% | 42.0% | 17.6% | 100.0% |
| Poor workflow planning and development | Count | 2 | 4 | 23 | 113 | 63 | 205 |
| | Row N% | 1.0% | 2.0% | 11.2% | 55.1% | 30.7% | 100.0% |
| Project overall failure | Count | 2 | 6 | 24 | 101 | 72 | 205 |
| | Row N% | 1.0% | 2.9% | 11.7% | 49.3% | 35.1% | 100.0% |
| Poor quality | Count | 1 | 5 | 43 | 118 | 38 | 205 |
| | Row N% | 0.5% | 2.4% | 21.0% | 57.6% | 18.5% | 100.0% |
| Additional costs | Count | 2 | 1 | 21 | 102 | 79 | 205 |
| | Row N% | 1.0% | 0.5% | 10.2% | 49.8% | 38.5% | 100.0% |
| Poor safety standards | Count | 1 | 5 | 35 | 116 | 48 | 205 |
| | Row N% | 0.5% | 2.4% | 17.1% | 56.6% | 23.4% | 100.0% |
| Extension of project delivery time | Count | 1 | 5 | 21 | 109 | 69 | 205 |
| | Row N% | 0.5% | 2.4% | 10.2% | 53.2% | 33.7% | 100.0% |

As can be seen in Table 11 above, project delays, the extension of project delivery time, poor safety standards, stakeholders' complaints, project overall failure, poor workflow planning and development, the loss of profit, additional costs, the reputational damage of an organisation, and claims for damage were identified as the major consequences of a poor and ineffective interface risk management approach in construction projects according to the responses received. Table 12 below shows the responses received for research objective 2—the extent to which interface risk management approaches influence project goals and objectives and the successful execution of construction projects in South Africa. The respondents rated their answers using the extent scale.

As shown in Table 12 above, alliancing and partnering agreements, the identification of construction supply chain risks during interface establishment, conflict resolution carried out by parties involved, clash detection as an integral part of the construction process for interface risk management, interface risk management by all the parties involved, clash detection as an integral part of the design process for interface risk management, assessing third parties' dependencies to identify new interfaces, the identification of interface risks in the conceptualisation stage of a project, the identification of interface risks in the interface's establishment phases, the identification of interface risks in the execution stage, defining standard methods and procedures, establishing a building information modelling (BIM) volume strategy, and creating a virtual construction model during the construction phase were identified as the major interface risk management approaches that have the most impact on project goals and objectives and the successful execution of construction projects in South Africa. Table 13 below shows the responses received regarding to what extent the following are causes of interface risks in construction projects.

**Table 12.** Responses received for research objective 2 for B3—the extent to which interface risk management approaches influence project goals and objectives and the successful execution of construction projects in South Africa.

| Section B3 (B3.1–B3.24) | | To no Extent | To a Small Extent | To a Moderate Extent | To a Large Extent | To a Very Large Extent | Total |
|---|---|---|---|---|---|---|---|
| Alliancing and partnering agreements | Count | 0 | 3 | 23 | 105 | 74 | 205 |
| | Row N% | 0.0% | 1.5% | 11.2% | 51.2% | 36.1% | 100.0% |
| Identifying third parties' dependencies to identify new interfaces | Count | 0 | 8 | 46 | 108 | 43 | 205 |
| | Row N% | 0.0% | 3.9% | 22.4% | 52.7% | 21.0% | 100.0% |
| Assessing third parties' dependencies to identify new interfaces | Count | 0 | 15 | 41 | 97 | 52 | 205 |
| | Row N% | 0.0% | 7.3% | 20.0% | 47.3% | 25.4% | 100.0% |
| Identifying third parties' dependencies to manage new interfaces | Count | 1 | 6 | 56 | 101 | 41 | 205 |
| | Row N% | 0.5% | 2.9% | 27.3% | 49.3% | 20.0% | 100.0% |
| Assessing third parties' dependencies to manage new interfaces | Count | 0 | 14 | 44 | 87 | 60 | 205 |
| | Row N% | 0.0% | 6.8% | 21.5% | 42.4% | 29.3% | 100.0% |
| Defining standard methods and procedures | Count | 1 | 6 | 26 | 102 | 70 | 205 |
| | Row N% | 0.5% | 2.9% | 12.7% | 49.8% | 34.1% | 100.0% |
| Establishing a building information modelling (BIM) volume strategy | Count | 0 | 7 | 23 | 92 | 83 | 205 |
| | Row N% | 0.0% | 3.4% | 11.2% | 44.9% | 40.5% | 100.0% |
| Creating a virtual construction model during the construction phase | Count | 2 | 9 | 23 | 104 | 67 | 205 |
| | Row N% | 1.0% | 4.4% | 11.2% | 50.7% | 32.7% | 100.0% |
| Regular meetings between project stakeholders | Count | 0 | 8 | 29 | 98 | 70 | 205 |
| | Row N% | 0.0% | 3.9% | 14.1% | 47.8% | 34.1% | 100.0% |
| Identification of construction supply chain risks during interface | Count | 1 | 4 | 19 | 119 | 62 | 205 |
| | Row N% | 0.5% | 2.0% | 9.3% | 58.0% | 30.2% | 100.0% |
| Identification of interface risks in the conceptualisation stage of a project | Count | 0 | 7 | 22 | 93 | 83 | 205 |
| | Row N% | 0.0% | 3.4% | 10.7% | 45.4% | 40.5% | 100.0% |
| Identification of interface risks in the planning stage of a project | Count | 0 | 4 | 17 | 108 | 76 | 205 |
| | Row N% | 0.0% | 2.0% | 8.3% | 52.7% | 37.1% | 100.0% |
| Identification of interface risks in the execution stage of a project | Count | 1 | 4 | 42 | 89 | 69 | 205 |
| | Row N% | 0.5% | 2.0% | 20.5% | 43.4% | 33.7% | 100.0% |
| Identification of interface risks in the interface's establishment phases | Count | 0 | 5 | 24 | 117 | 59 | 205 |
| | Row N% | 0.0% | 2.4% | 11.7% | 57.1% | 28.8% | 100.0% |
| Identification of interface risks in the execution stage | Count | 0 | 7 | 40 | 88 | 70 | 205 |
| | Row N% | 0.0% | 3.4% | 19.5% | 42.9% | 34.1% | 100.0% |
| Stakeholders' management strategies to predict how the project will affect | Count | 0 | 8 | 30 | 133 | 34 | 205 |
| | Row N% | 0.0% | 3.9% | 14.6% | 64.9% | 16.6% | 100.0% |
| Stakeholders mapping to predict how stakeholders will affect the project | Count | 1 | 10 | 48 | 93 | 53 | 205 |
| | Row N% | 0.5% | 4.9% | 23.4% | 45.4% | 25.9% | 100.0% |
| Clash avoidance as an integral part of the construction process for interface | Count | 0 | 7 | 41 | 118 | 39 | 205 |
| | Row N% | 0.0% | 3.4% | 20.0% | 57.6% | 19.0% | 100.0% |
| Clash avoidance as an integral part of the design process for interface risk | Count | 0 | 13 | 55 | 88 | 49 | 205 |
| | Row N% | 0.0% | 6.3% | 26.8% | 42.9% | 23.9% | 100.0% |
| Clash detection as an integral part of the construction process for interface risk | Count | 0 | 8 | 47 | 112 | 38 | 205 |
| | Row N% | 0.0% | 3.9% | 22.9% | 54.6% | 18.5% | 100.0% |
| Clash detection as an integral part of the design process for interface risk | Count | 0 | 9 | 47 | 98 | 51 | 205 |
| | Row N% | 0.0% | 4.4% | 22.9% | 47.8% | 24.9% | 100.0% |
| Conflicts resolution carried out by parties involved | Count | 0 | 4 | 22 | 112 | 67 | 205 |
| | Row N% | 0.0% | 2.0% | 10.7% | 54.6% | 32.7% | 100.0% |
| Collaboration between project stakeholders | Count | 1 | 4 | 19 | 82 | 99 | 205 |
| | Row N% | 0.5% | 2.0% | 9.3% | 40.0% | 48.3% | 100.0% |
| Interface risk management by all the parties involved | Count | 0 | 4 | 15 | 102 | 84 | 205 |
| | Row N% | 0.0% | 2.0% | 7.3% | 49.8% | 41.0% | 100.0% |

As can be seen in Table 13 above, the responses indicated that disorganized construction supply chain management, incompetency, poor workflow planning and development, subcontractors' negative attitudes towards teamwork, unpredictable and low delivery

reliability, poor inventories, a lack of knowledge sharing, procurement delays, ineffective communication in site layout changes with stakeholders, a poor understanding of the construction project process among project stakeholders, not updating changes in site layout with stakeholders, and disorganized construction supply chain management were identified as the major causes of interface risks in construction projects.

**Table 13.** Responses to research objective 3 for B4 (To what extent are the following the causes of interface risks on construction projects?).

| Section B4 (B4.1–B4) | | To no Extent | To a Small Extent | To a Moderate Extent | To a Large Extent | To a Very Large Extent | Total |
|---|---|---|---|---|---|---|---|
| Poor workflow planning and development | Count | 1 | 3 | 11 | 110 | 80 | 205 |
| | Row N% | 0.5% | 1.5% | 5.4% | 53.7% | 39.0% | 100.0% |
| Subcontractors' negative attitudes towards teamwork | Count | 1 | 3 | 39 | 114 | 48 | 205 |
| | Row N% | 0.5% | 1.5% | 19.0% | 55.6% | 23.4% | 100.0% |
| Procurement delays | Count | 1 | 4 | 40 | 104 | 56 | 205 |
| | Row N% | 0.5% | 2.0% | 19.5% | 50.7% | 27.3% | 100.0% |
| Unpredictable and low delivery reliability | Count | 1 | 6 | 37 | 115 | 46 | 205 |
| | Row N% | 0.5% | 2.9% | 18.0% | 56.1% | 22.4% | 100.0% |
| Poor inventories | Count | 1 | 8 | 37 | 102 | 57 | 205 |
| | Row N% | 0.5% | 3.9% | 18.0% | 49.8% | 27.8% | 100.0% |
| Lack of knowledge sharing | Count | 0 | 5 | 22 | 95 | 83 | 205 |
| | Row N% | 0.0% | 2.4% | 10.7% | 46.3% | 40.5% | 100.0% |
| Poor understanding of the construction project process among | Count | 1 | 5 | 17 | 106 | 76 | 205 |
| | Row N% | 0.5% | 2.4% | 8.3% | 51.7% | 37.1% | 100.0% |
| Not updating changes in site layout with stakeholders | Count | 1 | 5 | 37 | 109 | 53 | 205 |
| | Row N% | 0.5% | 2.4% | 18.0% | 53.2% | 25.9% | 100.0% |
| Ineffective communication in site layout changes with stakeholders | Count | 1 | 3 | 19 | 82 | 100 | 205 |
| | Row N% | 0.5% | 1.5% | 9.3% | 40.0% | 48.8% | 100.0% |
| Disorganized construction supply chain management | Count | 1 | 4 | 17 | 105 | 78 | 205 |
| | Row N% | 0.5% | 2.0% | 8.3% | 51.2% | 38.0% | 100.0% |
| Neglecting the handover process between two activities involving | Count | 0 | 7 | 46 | 105 | 47 | 205 |
| | Row N% | 0.0% | 3.4% | 22.4% | 51.2% | 22.9% | 100.0% |
| Excluding subcontractors during the planning stage of a project | Count | 4 | 5 | 52 | 106 | 38 | 205 |
| | Row N% | 2.0% | 2.4% | 25.4% | 51.7% | 18.5% | 100.0% |
| Clients' negative attitudes toward project stakeholders | Count | 3 | 4 | 36 | 101 | 61 | 205 |
| | Row N% | 1.5% | 2.0% | 17.6% | 49.3% | 29.8% | 100.0% |
| Incompetency | Count | 0 | 3 | 22 | 101 | 79 | 205 |
| | Row N% | 0.0% | 1.5% | 10.7% | 49.3% | 38.5% | 100.0% |
| Absence of contractors in project coordination meetings | Count | 1 | 7 | 33 | 108 | 56 | 205 |
| | Row N% | 0.5% | 3.4% | 16.1% | 52.7% | 27.3% | 100.0% |
| Absence of subcontractors in project coordination meetings | Count | 0 | 7 | 39 | 118 | 41 | 205 |
| | Row N% | 0.0% | 3.4% | 19.0% | 57.6% | 20.0% | 100.0% |
| Absence of suppliers and vendors in project coordination meetings | Count | 2 | 18 | 73 | 72 | 40 | 205 |
| | Row N% | 1.0% | 8.8% | 35.6% | 35.1% | 19.5% | 100.0% |
| Absence of vendors in project coordination meetings | Count | 3 | 25 | 69 | 84 | 24 | 205 |
| | Row N% | 1.5% | 12.2% | 33.7% | 41.0% | 11.7% | 100.0% |
| Contractors' negative attitudes toward project stakeholders | Count | 0 | 8 | 39 | 112 | 46 | 205 |
| | Row N% | 0.0% | 3.9% | 19.0% | 54.6% | 22.4% | 100.0% |

## 4.1. Exploratory Factor Analysis

Since the sample size was 205, exploratory factor analysis was performed to reduce the data or summarise using a smaller set of factors or components. This was achieved by looking for groups among the intercorrelations of the set of variables. By using factor analytic techniques, data were refined and reduced to form a smaller number of related variables to a more manageable number before using them in other analyses. The factorability of the correlation matrix was as follows: to be considered suitable for factor analysis, the

correlation matrix should show at least have some correlations of r = 0.3 or greater. Barlett's test of sphericity should be statistically significant at $p < 0.05$, and the Kaiser–Meyer–Olkin values should be 0.6 or above. These values are presented as part of the output of the factor analysis. Table 14 below depicts the exploratory factor analysis for research objective 1.

**Table 14.** Exploratory factor analysis for research objective 1–B2 (The consequences of poor and ineffective interface risk management approach).

| Factor | Initial Eigenvalues | | | Extraction Sums of Squared Loadings | | | Rotation Sums of Squared Loadings | | |
|---|---|---|---|---|---|---|---|---|---|
| | Total | % of Variance | Cumulative % | Total | % of Variance | Cumulative % | Total | % of Variance | Cumulative % |
| 1 | 6.206 | 47.742 | 47.742 | 5.732 | 44.093 | 44.093 | 3.373 | 25.946 | 25.946 |
| 2 | 1.438 | 11.063 | **58.805** | 0.973 | 7.483 | 51.576 | 3.332 | 25.630 | **51.576** |
| 3 | 0.917 | 7.053 | 65.858 | | | | | | |
| 4 | 0.715 | 5.498 | 71.355 | | | | | | |
| 5 | 0.593 | 4.559 | 75.914 | | | | | | |
| 6 | 0.536 | 4.120 | 80.034 | | | | | | |
| 7 | 0.503 | 3.868 | 83.903 | | | | | | |
| 8 | 0.454 | 3.493 | 87.395 | | | | | | |
| 9 | 0.422 | 3.246 | 90.641 | | | | | | |
| 10 | 0.402 | 3.091 | 93.732 | | | | | | |
| 11 | 0.336 | 2.587 | 96.318 | | | | | | |
| 12 | 0.253 | 1.943 | 98.261 | | | | | | |
| 13 | 0.226 | 1.739 | 100.000 | | | | | | |

As shown in Table 14 above, the consequences of poor and ineffective interface risk management approaches were loaded on two factors with eigenvalues of 6.206 and 1.438. These two factors explained 58.805% of the variance before rotation and 51.576% of the variance after rotation, and they represent the major and minor consequences of poor and ineffective interface risk management approaches. Table 15 below shows the corelation matrix for research objective 1.

**Table 15.** Correlation matrix for research objective 1–B2 (The consequences of poor and ineffective interface risk management approaches.).

| | | | | | | | Correlation Matrix | | | | | | |
|---|---|---|---|---|---|---|---|---|---|---|---|---|---|
| | | B2.1 | B2.2 | B2.3 | B2.4 | B2.5 | B2.6 | B2.7 | B2.8 | B2.9 | B2.10 | B2.11 | B2.12 | B2.13 |
| Correlation | B2.1 | 1.000 | 0.267 | 0.349 | 0.360 | 0.487 | 0.288 | 0.465 | 0.328 | 0.444 | 0.265 | 0.438 | 0.361 | 0.491 |
| | B2.2 | 0.267 | 1.000 | 0.438 | 0.552 | 0.259 | 0.542 | 0.374 | 0.487 | 0.328 | 0.501 | 0.298 | 0.448 | 0.339 |
| | B2.3 | 0.349 | 0.438 | 1.000 | 0.408 | 0.310 | 0.453 | 0.474 | 0.321 | 0.523 | 0.347 | 0.568 | 0.384 | 0.471 |
| | B2.4 | 0.360 | 0.552 | 0.408 | 1.000 | 0.395 | 0.508 | 0.414 | 0.545 | 0.471 | 0.594 | 0.413 | 0.504 | 0.342 |
| | B2.5 | 0.487 | 0.259 | 0.310 | 0.395 | 1.000 | 0.275 | 0.516 | 0.376 | 0.442 | 0.426 | 0.384 | 0.337 | 0.418 |
| | B2.6 | 0.288 | 0.542 | 0.453 | 0.508 | 0.275 | 1.000 | 0.268 | 0.641 | 0.431 | 0.590 | 0.534 | 0.511 | 0.397 |
| | B2.7 | 0.465 | 0.374 | 0.474 | 0.414 | 0.516 | 0.268 | 1.000 | 0.354 | 0.504 | 0.411 | 0.485 | 0.403 | 0.429 |
| | B2.8 | 0.328 | 0.487 | 0.321 | 0.545 | 0.376 | 0.641 | 0.354 | 1.000 | 0.378 | 0.616 | 0.430 | 0.586 | 0.379 |
| | B2.9 | 0.444 | 0.328 | 0.523 | 0.471 | 0.442 | 0.431 | 0.504 | 0.378 | 1.000 | 0.371 | 0.526 | 0.401 | 0.569 |
| | B2.10 | 0.265 | 0.501 | 0.347 | 0.594 | 0.426 | 0.590 | 0.411 | 0.616 | 0.371 | 1.000 | 0.407 | 0.672 | 0.317 |
| | B2.11 | 0.438 | 0.298 | 0.568 | 0.413 | 0.384 | 0.534 | 0.485 | 0.430 | 0.526 | 0.407 | 1.000 | 0.388 | 0.624 |
| | B2.12 | 0.361 | 0.448 | 0.384 | 0.504 | 0.337 | 0.511 | 0.403 | 0.586 | 0.401 | 0.672 | 0.388 | 1.000 | 0.364 |
| | B2.13 | 0.491 | 0.339 | 0.471 | 0.342 | 0.418 | 0.397 | 0.429 | 0.379 | 0.569 | 0.317 | 0.624 | 0.364 | 1.000 |

As can be seen in Table 15 above, many of the correlations were greater than 0.3. B2.2 and B2.1 had a correlation of 0.267, B2.6 and B2.1 had a correlation of 0.288, B2.11 and B2.2 had a correlation of 0.298, B2.2 and B2.5 had a correlation of 0.259, and B2.1 and B2.10 had a correlation of 0.265. Table 16 below shows the communalities for research objective 1–B2.

**Table 16.** The communalities for objective 1–B2.

| Communalities | | |
| --- | --- | --- |
| **Section B2.1–B2.13** | **Initial** | **Extraction** |
| B2.1 | 0.400 | 0.416 |
| B2.2 | 0.475 | 0.436 |
| B2.3 | 0.472 | 0.437 |
| B2.4 | 0.516 | 0.531 |
| B2.5 | 0.421 | 0.368 |
| B2.6 | 0.604 | 0.568 |
| B2.7 | 0.484 | 0.470 |
| B2.8 | 0.559 | 0.602 |
| B2.9 | 0.502 | 0.549 |
| B2.10 | 0.626 | 0.680 |
| B2.11 | 0.582 | 0.555 |
| B2.12 | 0.538 | 0.539 |
| B2.13 | 0.519 | 0.553 |

Extraction method: principal axis factoring.

As shown in Table 16 above, the extractions were all above 0.3, which means there was at least a 30% common variance shared among the items in Section B2. Figure 1 below shows the scree plot for Section B2.

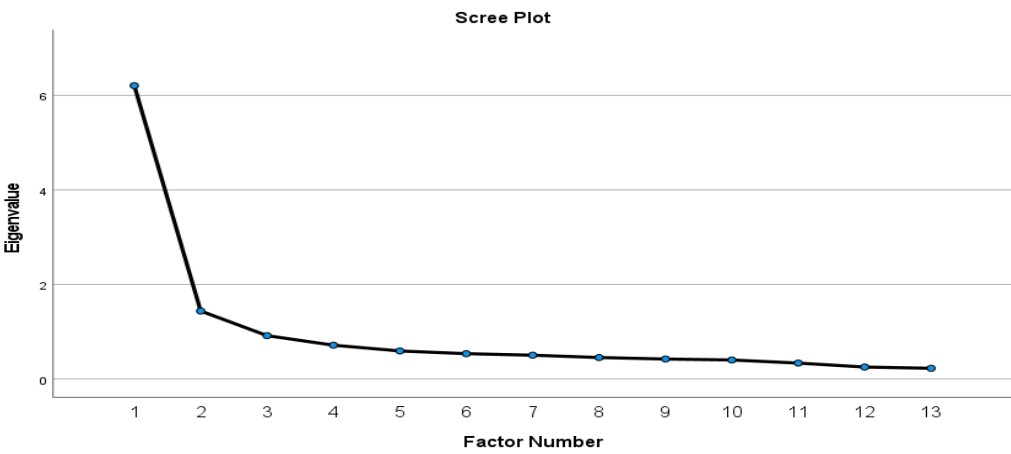

**Figure 1.** Scree plot for research objective 1–B2.

In Figure 1 above, variables 3 to 13 were not significant, so they were not added to the factor rotation. For the first factor, the eigenvalue was around 6.2; for the second factor, the eigenvalue was around 1.4. Table 17 below depicts the rotated factor matrix for research objective 1–B2.

Table 17 shows that these items were grouped into two factors based on the ten key performance factors (KPIs) that are related to construction projects, namely time, cost, people, quality, safety and health, internal and external stakeholder, client satisfaction, financial performance, environment, information, and technology and innovation. Factor 1 items are associated with quality, people, IT and innovations, time, cost, safety and health, and environment and client satisfaction, while items in factor 2 items are associated with cost, time, external and internal stakeholders, financial performance, people, environment, and client satisfaction. Table 18 below represents the exploratory factor analysis for research objective 2 ("What are the interface risk management approaches by organisations?")

**Table 17.** Rotated factor matrix for research objective 1–B2.

| | Rotated Factor Matrix [a] | |
|---|---|---|
| | Factor | |
| | 1 | 2 |
| B2.10 Poor quality | **0.791** | 0.233 |
| B2.8 Poor workflow planning and development | **0.727** | 0.271 |
| B2.6 Project delays | **0.689** | 0.305 |
| B2.12 Poor safety standards | **0.668** | 0.305 |
| B2.4 Reputational damage of an organisation | **0.634** | 0.359 |
| B2.2 Claims for damage | **0.606** | 0.264 |
| B2.13 Extension of project delivery time | 0.220 | **0.711** |
| B2.9 Project overall failure | 0.289 | **0.682** |
| B2.11 Additional costs | 0.319 | **0.674** |
| B2.7 Regulatory infringements | 0.281 | **0.625** |
| B2.1 Stakeholders' complaints | 0.191 | **0.616** |
| B2.3 Loss of profit | 0.324 | **0.576** |
| B2.5 Industrial actions | 0.275 | **0.541** |

Extraction method: principal axis factoring. Rotation method: varimax with Kaiser normalization. [a]. Rotation converged in three iterations.

**Table 18.** Exploratory factor analysis for research objective 2 ("Interface risk management approaches by organisations.").

| Factor | Initial Eigenvalues | | | Extraction Sums of Squared Loadings | | | Rotation Sums of Squared Loadings | | |
|---|---|---|---|---|---|---|---|---|---|
| | Total | % of Variance | Cumulative % | Total | % of Variance | Cumulative % | Total | % of Variance | Cumulative % |
| 1 | 11.460 | 47.748 | 47.748 | 11.120 | 46.333 | 46.333 | 6.373 | 26.553 | 26.553 |
| 2 | 2.787 | 11.613 | 59.361 | 2.456 | 10.235 | 56.569 | 4.029 | 16.786 | 43.339 |
| 3 | 1.581 | 6.589 | 65.950 | 1.272 | 5.298 | 61.867 | 3.740 | 15.582 | 58.922 |
| 4 | 1.209 | 5.037 | **70.987** | 0.844 | 3.517 | 65.383 | 1.551 | 6.462 | **65.383** |
| 5 | 0.947 | 3.944 | 74.931 | | | | | | |
| 6 | 0.741 | 3.089 | 78.021 | | | | | | |
| 7 | 0.590 | 2.459 | 80.479 | | | | | | |
| 8 | 0.522 | 2.174 | 82.653 | | | | | | |
| 9 | 0.494 | 2.057 | 84.710 | | | | | | |
| 10 | 0.479 | 1.994 | 86.704 | | | | | | |
| 11 | 0.445 | 1.856 | 88.560 | | | | | | |
| 12 | 0.376 | 1.568 | 90.128 | | | | | | |
| 13 | 0.318 | 1.325 | 91.452 | | | | | | |
| 14 | 0.290 | 1.210 | 92.662 | | | | | | |
| 15 | 0.258 | 1.075 | 93.738 | | | | | | |
| 16 | 0.233 | 0.970 | 94.708 | | | | | | |
| 17 | 0.210 | 0.877 | 95.585 | | | | | | |
| 18 | 0.200 | 0.835 | 96.420 | | | | | | |
| 19 | 0.188 | 0.785 | 97.204 | | | | | | |
| 20 | 0.170 | 0.707 | 97.911 | | | | | | |
| 21 | 0.151 | 0.631 | 98.542 | | | | | | |
| 22 | 0.141 | 0.587 | 99.129 | | | | | | |
| 23 | 0.122 | 0.509 | 99.637 | | | | | | |
| 24 | 0.087 | 0.363 | 100.000 | | | | | | |

As seen in Table 18 above, "Interface risk management approaches by organisation" the values were loaded on four factors with eigenvalues of 11.460, 2.787, 1.581, and 1.209. These four factors explained 70.987% of the variance before rotation and 65,383% of the variance after rotation. Table 19 below shows the correlation matrices for research objective 2–B3.

**Table 19.** Correlation matrices for Section B3.

| | B3.1 | B3.2 | B3.3 | B3.4 | B3.5 | B3.6 | B3.7 | B3.8 | B3.9 | B3.10 | B3.11 | B3.12 | B3.13 | B3.14 | B3.15 | B3.16 | B3.17 | B3.18 | B3.19 | B3.20 | B3.21 | B3.22 | B3.23 | B3.24 |
|---|---|---|---|---|---|---|---|---|---|---|---|---|---|---|---|---|---|---|---|---|---|---|---|---|
| B3.1 | 1.000 | 0.370 | 0.581 | 0.265 | 0.483 | 0.266 | 0.559 | 0.258 | 0.537 | 0.380 | 0.560 | 0.285 | 0.480 | 0.397 | 0.528 | 0.346 | 0.428 | 0.316 | 0.523 | 0.372 | 0.488 | 0.394 | 0.476 | 0.353 |
| B3.2 | 0.370 | 1.000 | 0.434 | 0.705 | 0.449 | 0.497 | 0.381 | 0.468 | 0.286 | 0.464 | 0.334 | 0.455 | 0.264 | 0.507 | 0.254 | 0.477 | 0.348 | 0.465 | 0.351 | 0.443 | 0.355 | 0.447 | 0.263 | 0.425 |
| B3.3 | 0.581 | 0.434 | 1.000 | 0.415 | 0.714 | 0.362 | 0.536 | 0.340 | 0.455 | 0.307 | 0.495 | 0.296 | 0.466 | 0.336 | 0.560 | 0.374 | 0.570 | 0.366 | 0.576 | 0.378 | 0.548 | 0.332 | 0.582 | 0.295 |
| B3.4 | 0.265 | 0.705 | 0.415 | 1.000 | 0.468 | 0.585 | 0.367 | 0.494 | 0.376 | 0.503 | 0.298 | 0.504 | 0.287 | 0.506 | 0.193 | 0.478 | 0.413 | 0.489 | 0.351 | 0.429 | 0.348 | 0.382 | 0.286 | 0.398 |
| B3.5 | 0.483 | 0.449 | 0.714 | 0.468 | 1.000 | 0.423 | 0.561 | 0.387 | 0.536 | 0.385 | 0.572 | 0.347 | 0.532 | 0.416 | 0.508 | 0.384 | 0.591 | 0.384 | 0.568 | 0.390 | 0.592 | 0.265 | 0.507 | 0.238 |
| B3.6 | 0.266 | 0.497 | 0.362 | 0.585 | 0.423 | 1.000 | 0.430 | 0.639 | 0.455 | 0.690 | 0.424 | 0.636 | 0.414 | 0.640 | 0.296 | 0.511 | 0.335 | 0.401 | 0.288 | 0.424 | 0.310 | 0.431 | 0.362 | 0.513 |
| B3.7 | 0.559 | 0.381 | 0.536 | 0.367 | 0.561 | 0.430 | 1.000 | 0.585 | 0.536 | 0.419 | 0.654 | 0.381 | 0.561 | 0.426 | 0.511 | 0.356 | 0.480 | 0.310 | 0.462 | 0.292 | 0.477 | 0.278 | 0.522 | 0.277 |
| B3.8 | 0.258 | 0.468 | 0.340 | 0.494 | 0.387 | 0.639 | 0.585 | 1.000 | 0.316 | 0.650 | 0.398 | 0.547 | 0.401 | 0.592 | 0.283 | 0.474 | 0.260 | 0.394 | 0.212 | 0.318 | 0.214 | 0.418 | 0.286 | 0.376 |
| B3.9 | 0.537 | 0.286 | 0.455 | 0.376 | 0.536 | 0.455 | 0.536 | 0.316 | 1.000 | 0.485 | 0.658 | 0.430 | 0.655 | 0.415 | 0.530 | 0.357 | 0.560 | 0.325 | 0.532 | 0.332 | 0.504 | 0.423 | 0.586 | 0.382 |
| B3.10 | 0.380 | 0.464 | 0.307 | 0.503 | 0.385 | 0.690 | 0.419 | 0.650 | 0.485 | 1.000 | 0.501 | 0.741 | 0.450 | 0.737 | 0.328 | 0.590 | 0.350 | 0.506 | 0.291 | 0.456 | 0.312 | 0.544 | 0.375 | 0.490 |
| B3.11 | 0.560 | 0.334 | 0.495 | 0.298 | 0.572 | 0.424 | 0.654 | 0.398 | 0.658 | 0.501 | 1.000 | 0.565 | 0.688 | 0.464 | 0.652 | 0.350 | 0.602 | 0.295 | 0.591 | 0.371 | 0.590 | 0.342 | 0.630 | 0.368 |
| B.12 | 0.285 | 0.455 | 0.296 | 0.504 | 0.347 | 0.636 | 0.381 | 0.547 | 0.430 | 0.741 | 0.565 | 1.000 | 0.378 | 0.721 | 0.331 | 0.543 | 0.322 | 0.483 | 0.306 | 0.479 | 0.305 | 0.532 | 0.408 | 0.537 |
| B3.13 | 0.480 | 0.264 | 0.466 | 0.287 | 0.532 | 0.414 | 0.561 | 0.401 | 0.655 | 0.450 | 0.688 | 0.378 | 1.000 | 0.483 | 0.757 | 0.397 | 0.590 | 0.352 | 0.564 | 0.330 | 0.509 | 0.296 | 0.509 | 0.221 |
| B3.14 | 0.397 | 0.507 | 0.336 | 0.506 | 0.416 | 0.640 | 0.426 | 0.592 | 0.415 | 0.737 | 0.464 | 0.721 | 0.483 | 1.000 | 0.402 | 0.691 | 0.396 | 0.580 | 0.382 | 0.544 | 0.353 | 0.610 | 0.389 | 0.544 |
| B3.15 | 0.528 | 0.254 | 0.560 | 0.193 | 0.508 | 0.296 | 0.511 | 0.283 | 0.530 | 0.328 | 0.652 | 0.331 | 0.757 | 0.402 | 1.000 | 0.376 | 0.599 | 0.325 | 0.616 | 0.344 | 0.579 | 0.303 | 0.544 | 0.237 |
| B3.16 | 0.346 | 0.477 | 0.374 | 0.478 | 0.384 | 0.511 | 0.356 | 0.474 | 0.357 | 0.590 | 0.350 | 0.543 | 0.397 | 0.691 | 0.376 | 1.000 | 0.529 | 0.724 | 0.501 | 0.648 | 0.419 | 0.591 | 0.356 | 0.464 |
| B3.17 | 0.428 | 0.348 | 0.570 | 0.413 | 0.591 | 0.335 | 0.480 | 0.260 | 0.560 | 0.350 | 0.602 | 0.322 | 0.590 | 0.396 | 0.599 | 0.529 | 1.000 | 0.450 | 0.830 | 0.444 | 0.795 | 0.358 | 0.585 | 0.253 |
| B3.18 | 0.316 | 0.465 | 0.366 | 0.489 | 0.384 | 0.401 | 0.310 | 0.394 | 0.325 | 0.506 | 0.295 | 0.483 | 0.352 | 0.580 | 0.325 | 0.724 | 0.450 | 1.000 | 0.555 | 0.798 | 0.419 | 0.624 | 0.357 | 0.539 |
| B3.19 | 0.523 | 0.351 | 0.576 | 0.351 | 0.568 | 0.288 | 0.462 | 0.212 | 0.532 | 0.291 | 0.591 | 0.306 | 0.564 | 0.382 | 0.616 | 0.501 | 0.830 | 0.555 | 1.000 | 0.534 | 0.852 | 0.400 | 0.593 | 0.310 |
| B3.20 | 0.372 | 0.443 | 0.378 | 0.429 | 0.390 | 0.424 | 0.292 | 0.318 | 0.332 | 0.456 | 0.371 | 0.479 | 0.330 | 0.544 | 0.344 | 0.648 | 0.444 | 0.798 | 0.534 | 1.000 | 0.465 | 0.619 | 0.363 | 0.547 |
| B3.21 | 0.488 | 0.355 | 0.548 | 0.348 | 0.592 | 0.310 | 0.477 | 0.214 | 0.504 | 0.312 | 0.590 | 0.305 | 0.509 | 0.353 | 0.579 | 0.419 | 0.795 | 0.419 | 0.852 | 0.465 | 1.000 | 0.363 | 0.568 | 0.310 |
| B3.22 | 0.394 | 0.447 | 0.332 | 0.382 | 0.265 | 0.431 | 0.278 | 0.418 | 0.423 | 0.544 | 0.342 | 0.532 | 0.296 | 0.610 | 0.303 | 0.591 | 0.358 | 0.624 | 0.400 | 0.619 | 0.363 | 1.000 | 0.512 | 0.738 |
| B3.23 | 0.476 | 0.263 | 0.582 | 0.286 | 0.507 | 0.362 | 0.522 | 0.286 | 0.586 | 0.375 | 0.630 | 0.408 | 0.509 | 0.389 | 0.544 | 0.356 | 0.585 | 0.357 | 0.593 | 0.363 | 0.568 | 0.512 | 1.000 | 0.534 |
| B3.24 | 0.353 | 0.425 | 0.295 | 0.398 | 0.238 | 0.513 | 0.277 | 0.376 | 0.382 | 0.490 | 0.368 | 0.537 | 0.221 | 0.544 | 0.237 | 0.464 | 0.253 | 0.539 | 0.310 | 0.547 | 0.310 | 0.738 | 0.534 | 1.000 |

As shown in Table 19 above, many of the correlations were greater than 0.3. B3.13 and B3.24 had a correlation of 0.221, B3.2 and B3.9 had a correlation of 0.286, B3.5 and B3.22 had a correlation of 0.265, B3.4 and B3.1 had a correlation of 0.265, and B3.2 and B3.23 had a correlation of 0.263. Table 20 below shows the communalities for research objective 2–B3.

**Table 20.** Communalities for research objective 2–B3.

| Communalities | | |
|---|---|---|
| **B3.1–B3.24** | **Initial** | **Extraction** |
| B3.1 | 0.604 | 0.446 |
| B3.2 | 0.599 | 0.575 |
| B3.3 | 0.677 | 0.576 |
| B3.4 | 0.653 | 0.671 |
| B3.5 | 0.662 | 0.649 |
| B3.6 | 0.670 | 0.645 |
| B3.7 | 0.668 | 0.589 |
| B3.8 | 0.667 | 0.609 |
| B3.9 | 0.650 | 0.581 |
| B3.10 | 0.742 | 0.732 |
| B3.11 | 0.776 | 0.759 |
| B3.12 | 0.724 | 0.664 |
| B3.13 | 0.735 | 0.641 |
| B3.14 | 0.738 | 0.715 |
| B3.15 | 0.687 | 0.628 |
| B3.16 | 0.696 | 0.633 |
| B3.17 | 0.794 | 0.729 |
| B3.18 | 0.787 | 0.741 |
| B3.19 | 0.852 | 0.856 |
| B3.20 | 0.710 | 0.685 |
| B3.21 | 0.787 | 0.720 |
| B3.22 | 0.714 | 0.699 |
| B3.23 | 0.672 | 0.579 |
| B3.24 | 0.687 | 0.569 |

Extraction method: principal axis factoring.

Table 20 above shows that the extractions are all above 0.3, which means there was at least a 30% common variance shared between them. Figure 2 below shows the scree plot of research objective 2–B3.

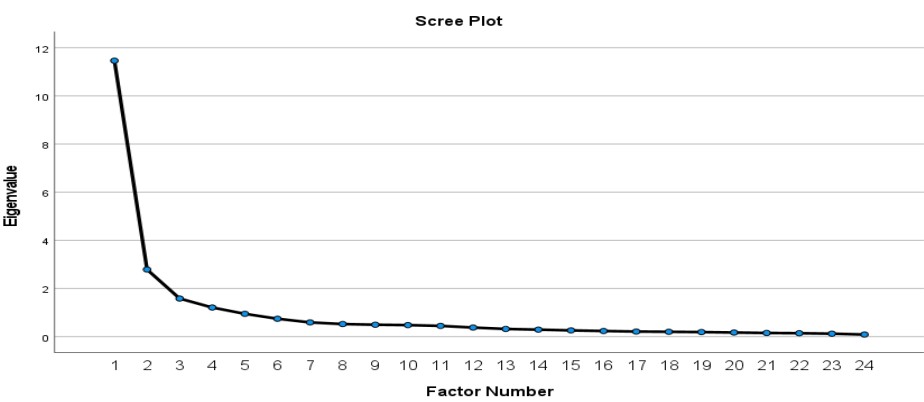

**Figure 2.** Scree plot of research objective 2–B3.

As can be seen in the scree plot above, variables 5 to 24 are not significant, so they were not added in the factor rotation. For the first factor, the eigenvalue was around 11.460; for the second factor, the eigenvalue was around 2.787; for the third factor, the eigenvalue was about 1.581; and for the fourth factor, the eigenvalue was 1.209. Table 21 below is the rotated factor matrix for Section B3.

**Table 21.** Rotated factor matrix for research objective 2–B3.

| | Factor [a] | | | |
|---|---|---|---|---|
| | **1** | **2** | **3** | **4** |
| B3.11 | **0.770** | 0.390 | 0.105 | −0.053 |
| B3.19 | **0.765** | −0.104 | 0.434 | 0.267 |
| B3.15 | **0.764** | 0.159 | 0.141 | −0.007 |
| B3.17 | **0.740** | 0.012 | 0.310 | 0.291 |
| B3.21 | **0.735** | −0.040 | 0.327 | 0.266 |
| B3.13 | **0.728** | 0.323 | 0.088 | −0.001 |
| B3.23 | **0.665** | 0.209 | 0.301 | −0.043 |
| B3.9 | **0.661** | 0.342 | 0.166 | −0.011 |
| B3.5 | **0.639** | 0.237 | 0.082 | 0.422 |
| B3.3 | **0.636** | 0.156 | 0.143 | 0.356 |
| B3.7 | **0.618** | 0.419 | 0.002 | 0.179 |
| B3.1 | **0.600** | 0.204 | 0.193 | 0.081 |
| B3.10 | 0.234 | **0.740** | 0.351 | 0.081 |
| B3.8 | 0.190 | **0.704** | 0.137 | 0.240 |
| B3.6 | 0.216 | **0.693** | 0.248 | 0.238 |
| B3.12 | 0.223 | **0.678** | 0.390 | 0.044 |
| B3.14 | 0.253 | **0.638** | 0.479 | 0.120 |
| B3.18 | 0.211 | 0.210 | **0.757** | 0.281 |
| B3.20 | 0.256 | 0.181 | **0.733** | 0.221 |
| B3.22 | 0.214 | 0.365 | **0.720** | −0.031 |
| B3.16 | 0.253 | 0.353 | **0.615** | 0.257 |
| B3.24 | 0.174 | 0.410 | **0.608** | −0.036 |
| B3.4 | 0.159 | 0.460 | 0.270 | **0.601** |
| B3.2 | 0.179 | 0.414 | 0.303 | **0.529** |

Extraction method: principal axis factoring. Rotation method: varimax with Kaiser normalization. [a]: rotation converged in nine iterations.

As seen in Table 21 above, B3.11, B3.19, B3.15, B3.17, B3.23, B3.9, B3.5, B3.7, B3.1, B3.13, and B3.21 were grouped into factor 1, while B3.8, B3.6, B3.10, B3.12, and B3.14 were grouped into factor 2 and B3.18, B3.20, B3.22, B3.16, and B3.24 were grouped into factor 3, while B3.4 and B3.2 were grouped into factor 4. These items were grouped into two factors based on the KPIs listed above; they are associated with as stated earlier. Factor 1 items are associated with quality, people, IT and innovations, time, financial performance, cost, safety and health, environment, internal and external stakeholders, and client satisfaction,

while items in factor 2 items are associated with cost, time, environment, quality, client satisfaction, information technology and innovation, external and internal stakeholders, people, environment and client satisfaction; factor 3 are items associated with environment, client satisfaction, safety and health, and external and internal stakeholders; and factor 4 are items associated with internal and external stakeholders. Table 22 below shows the exploratory factor analysis for research objective 3 ("The causes of interface risks").

**Table 22.** Total variance explained for research objective 3–B4 ("The causes of interface risks").

| Factor | Initial Eigenvalues | | | Extraction Sums of Squared Loadings | | | Rotation Sums of Squared Loadings | | |
|---|---|---|---|---|---|---|---|---|---|
| | Total | % of Variance | Cumulative % | Total | % of Variance | Cumulative % | Total | % of Variance | Cumulative % |
| 1 | 9.587 | 50.460 | 50.460 | 9.204 | 48.443 | 48.443 | 4.820 | 25.367 | 25.367 |
| 2 | 1.960 | 10.317 | 60.776 | 1.568 | 8.251 | 56.694 | 4.007 | 21.089 | 46.456 |
| 3 | 1.194 | 6.285 | 67.061 | 0.821 | 4.320 | 61.014 | 2.766 | 14.558 | **61.014** |
| 4 | 0.856 | 4.507 | 71.569 | | | | | | |
| 5 | 0.697 | 3.669 | 75.237 | | | | | | |
| 6 | 0.640 | 3.370 | 78.607 | | | | | | |
| 7 | 0.561 | 2.954 | 81.561 | | | | | | |
| 8 | 0.470 | 2.476 | 84.036 | | | | | | |
| 9 | 0.429 | 2.255 | 86.292 | | | | | | |
| 10 | 0.388 | 2.041 | 88.332 | | | | | | |
| 11 | 0.353 | 1.856 | 90.188 | | | | | | |
| 12 | 0.331 | 1.740 | 91.928 | | | | | | |
| 13 | 0.323 | 1.701 | 93.629 | | | | | | |
| 14 | 0.272 | 1.433 | 95.062 | | | | | | |
| 15 | 0.235 | 1.235 | 96.297 | | | | | | |
| 16 | 0.225 | 1.185 | 97.482 | | | | | | |
| 17 | 0.190 | 1.000 | 98.481 | | | | | | |
| 18 | 0.168 | 0.883 | 99.364 | | | | | | |
| 19 | 0.121 | 0.636 | 100.000 | | | | | | |

As shown in Table 22 above, the causes of interface risks were loaded on three factors with eigenvalues of 9.587, 1.960, and 1.194. These three factors explained 67.061% of the variance before rotation and 61.014% of the variance after rotation. Table 23 below shows the communalities for research objective 3.

Table 23 below depicts the communalities for research objective 3–B4.

As seen in Table 23 above, the extractions are all above 0.3, which means there was at least a 30% common variance shared among them. Figure 3 below shows the scree plot for research objective 3–B4.

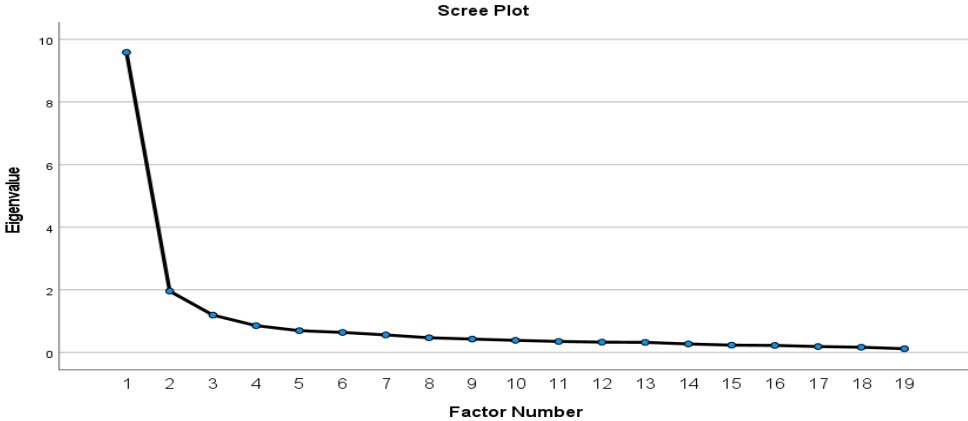

**Figure 3.** Scree plot for research objective 3–B4.

**Table 23.** Communalities for research objective 3–B4.

| | Communalities | |
|---|---|---|
| | **Initial** | **Extraction** |
| B4.1 | 0.551 | 0.556 |
| B4.2 | 0.560 | 0.500 |
| B4.3 | 0.625 | 0.578 |
| B4.4 | 0.624 | 0.542 |
| B4.5 | 0.612 | 0.598 |
| B4.6 | 0.588 | 0.588 |
| B4.7 | 0.679 | 0.719 |
| B4.8 | 0.624 | 0.569 |
| B4.9 | 0.617 | 0.598 |
| B4.10 | 0.658 | 0.727 |
| B4.11 | 0.650 | 0.605 |
| B4.12 | 0.702 | 0.719 |
| B4.13 | 0.621 | 0.541 |
| B4.14 | 0.572 | 0.515 |
| B4.15 | 0.704 | 0.663 |
| B4.16 | 0.672 | 0.649 |
| B4.17 | 0.786 | 0.687 |
| B4.18 | 0.732 | 0.591 |
| B4.19 | 0.679 | 0.647 |

Extraction method: principal axis factoring.

As can be seen in the scree plot above, variables 4 to 19 are not significant, so they were not added to the factor rotation. For the first factor, the eigenvalue was around 9.587; for the second factor, the eigenvalue was around 1.960; and for the third factor, the eigenvalue was about 1.194. Table 24 below is the rotated factor matrix for research objective 3–B4.

**Table 24.** Rotated factor matrix for research objective 3–B4.

| Rotated Factor Matrix [a] | | | |
|---|---|---|---|
| | **Factor** | | |
| | **1** | **2** | **3** |
| B4.7 Poor understanding of the construction project process among project stakeholders | **0.775** | 0.316 | 0.135 |
| B4.3 Procurement delays | **0.690** | 0.242 | 0.207 |
| B4.1 Poor workflow planning and development | **0.688** | 0.240 | 0.158 |
| B4.5 Poor inventories | **0.676** | 0.252 | 0.280 |
| B4.9 Ineffective communication in site layout changes with stakeholders | **0.669** | 0.386 | 0.040 |
| B4.11 Neglecting the handover process between two activities involving different trades in the planning stage | **0.659** | 0.208 | 0.357 |
| B4.15 Absence of contractors in project coordination meetings | **0.630** | 0.244 | 0.454 |
| B4.17 Absence of suppliers and vendors in project coordination meetings | **0.627** | 0.105 | 0.531 |
| B4.19 Contractors' negative attitudes toward project stakeholders | **0.580** | 0.195 | 0.521 |
| B4.10 Disorganized construction supply chain management | 0.255 | **0.805** | 0.121 |
| B4.6 Lack of knowledge sharing | 0.236 | **0.717** | 0.136 |
| B4.8 Not updating changes in site layout with stakeholders | 0.284 | **0.670** | 0.200 |
| B4.14 Incompetency | 0.191 | **0.660** | 0.206 |
| B4.4 Unpredictable and low delivery reliability | 0.325 | **0.635** | 0.180 |
| B4.2 Subcontractors' negative attitudes towards teamwork | 0.116 | **0.568** | 0.404 |
| B4.12 Excluding subcontractors during the planning stage of a project | 0.205 | 0.507 | **0.648** |
| B4.18 Absence of vendors in project coordination meetings | 0.350 | 0.332 | **0.598** |
| B4.16 Absence of subcontractors in project coordination meetings | 0.201 | 0.522 | **0.580** |
| B4.13 Clients' negative attitudes toward project stakeholders | 0.491 | 0.119 | **0.535** |

Extraction method: principal axis factoring. Rotation method: varimax with Kaiser normalization. [a]: rotation converged in 11 iterations.

As shown in Table 24 above, each item was grouped into factors based on the KPIs they are associated with, as stated earlier. B4.7, B4.1, B4.5, B4.9, B4.11, B4.15, B4.17, and B4.19 were grouped into factor 1; B4.10, B4.6, B4.8, B4.14, B4.4, and B4.2 were grouped into factor 2; and B4.18, B6.16, B4.13, and B4.12 were grouped into factor 3.

*4.2. Reliability Statistics of Collected Data*

4.2.1. Reliability Statistics of Theoretical Factors

To establish the consistency of the data, the value of the Cronbach's alpha (coefficient alpha) was determined. Table 25 below shows the reliability statistics for research objective 1, and Cronbach's alpha coefficients must be greater than 0.7 to confirm reliability and internal consistency.

**Table 25.** Reliability statistics for research objective 1–B2 ("The consequences of poor and ineffective interface risk management approaches.").

| Reliability Statistics | |
|---|---|
| **Cronbach's Alpha** | **N of Items** |
| 0.907 | 13 |

As shown in the above Table 25, Cronbach's alpha is 0.907, which was above 0.7; therefore, it is reliable. Table 26 below shows the item-total statistics for research objective 1.

**Table 26.** Item-total statistics for research objective 1.

| Item-Total Statistics | | | | |
|---|---|---|---|---|
| | **Scale Mean if Item Deleted** | **Scale Variance if Item Deleted** | **Corrected Item-Total Correlation** | **Cronbach's Alpha if Item Deleted** |
| B2.1 | 48.80 | 41.932 | 0.544 | 0.904 |
| B2.2 | 48.78 | 41.420 | 0.577 | 0.902 |
| B2.3 | 48.46 | 41.916 | 0.606 | 0.901 |
| B2.4 | 48.75 | 40.433 | 0.666 | 0.898 |
| B2.5 | 49.00 | 41.382 | 0.554 | 0.904 |
| B2.6 | 48.59 | 41.253 | 0.653 | 0.899 |
| B2.7 | 49.08 | 41.121 | 0.615 | 0.901 |
| B2.8 | 48.68 | 41.178 | 0.656 | 0.899 |
| B2.9 | 48.66 | 40.744 | 0.650 | 0.899 |
| B2.10 | 48.90 | 41.328 | 0.667 | 0.899 |
| B2.11 | 48.57 | 41.335 | 0.662 | 0.899 |
| B2.12 | 48.81 | 41.420 | 0.645 | 0.900 |
| B2.13 | 48.64 | 41.624 | 0.618 | 0.901 |

Table 26 above contains total statistics for all the items in B2 for research objective1.

Table 27 below shows a Cronbach's alpha value of 0.952, which was above 0.7; therefore, it is reliable. Table 28 below shows the item-total statistics for research objective 2.

**Table 27.** Reliability statistics for research objective 2–B3 ("What are the interface risk management approaches by organisation?").

| Reliability Statistics | |
|---|---|
| **Cronbach's Alpha** | **N of Items** |
| 0.952 | 24 |

Table 28 above contains total statistics for all the items in B3 for research objective 2. Table 29 below depicts the reliability statistics for research objective 3–B4.

**Table 28.** Item-total statistics for research objective 2.

| | Scale Mean if Item Deleted | Scale Variance if Item Deleted | Corrected Item-Total Correlation | Cronbach's Alpha if Item Deleted |
|---|---|---|---|---|
| | **Item-Total Statistics** | | | |
| B3.1 | 93.35 | 152.170 | 0.611 | 0.950 |
| B3.2 | 93.66 | 151.499 | 0.588 | 0.950 |
| B3.3 | 93.66 | 148.607 | 0.658 | 0.950 |
| B3.4 | 93.72 | 151.057 | 0.595 | 0.950 |
| B3.5 | 93.63 | 147.695 | 0.684 | 0.949 |
| B3.6 | 93.43 | 150.207 | 0.643 | 0.950 |
| B3.7 | 93.35 | 149.982 | 0.659 | 0.950 |
| B3.8 | 93.47 | 150.662 | 0.576 | 0.951 |
| B3.9 | 93.45 | 149.435 | 0.676 | 0.949 |
| B3.10 | 93.41 | 150.881 | 0.682 | 0.949 |
| B3.11 | 93.34 | 148.834 | 0.727 | 0.949 |
| B3.12 | 93.32 | 151.621 | 0.654 | 0.950 |
| B3.13 | 93.49 | 149.114 | 0.674 | 0.949 |
| B3.14 | 93.45 | 150.327 | 0.719 | 0.949 |
| B3.15 | 93.49 | 149.653 | 0.641 | 0.950 |
| B3.16 | 93.63 | 151.205 | 0.683 | 0.949 |
| B3.17 | 93.66 | 147.716 | 0.710 | 0.949 |
| B3.18 | 93.65 | 151.062 | 0.651 | 0.950 |
| B3.19 | 93.73 | 147.484 | 0.715 | 0.949 |
| B3.20 | 93.69 | 150.822 | 0.642 | 0.950 |
| B3.21 | 93.64 | 149.036 | 0.683 | 0.949 |
| B3.22 | 93.39 | 151.896 | 0.630 | 0.950 |
| B3.23 | 93.23 | 149.965 | 0.672 | 0.950 |
| B3.24 | 93.27 | 152.896 | 0.574 | 0.951 |

**Table 29.** Reliability statistics for research objective 3–B4 ("The causes of interface risks.").

| Cronbach's Alpha | N of Items |
|---|---|
| **Reliability Statistics** | |
| 0.945 | 19 |

As seen in Table 29 above, Cronbach's alpha is 0.945; therefore, it is reliable. Table 30 below shows the item-total statistics for research objective 3–B4 ("The causes of interface risks.").

**Table 30.** Item-total statistics for research objective 3–B4 ("The causes of interface risks.").

| | Scale Mean if Item Deleted | Scale Variance if Item Deleted | Corrected Item-Total Correlation | Cronbach's Alpha if Item Deleted |
|---|---|---|---|---|
| | **Item-Total Statistics** | | | |
| B4.1 | 72.19 | 99.701 | 0.645 | 0.942 |
| B4.2 | 72.48 | 100.006 | 0.577 | 0.943 |
| B4.3 | 72.45 | 98.082 | 0.672 | 0.942 |
| B4.4 | 72.51 | 98.683 | 0.646 | 0.942 |
| B4.5 | 72.47 | 96.966 | 0.705 | 0.941 |
| B4.6 | 72.23 | 99.413 | 0.606 | 0.943 |
| B4.7 | 72.25 | 97.700 | 0.730 | 0.941 |
| B4.8 | 72.46 | 98.583 | 0.644 | 0.942 |
| B4.9 | 72.13 | 98.631 | 0.654 | 0.942 |
| B4.10 | 72.23 | 98.945 | 0.654 | 0.942 |
| B4.11 | 72.54 | 97.642 | 0.705 | 0.941 |
| B4.12 | 72.65 | 96.659 | 0.712 | 0.941 |
| B4.13 | 72.44 | 97.973 | 0.627 | 0.943 |
| B4.14 | 72.23 | 100.315 | 0.578 | 0.943 |
| B4.15 | 72.45 | 96.690 | 0.752 | 0.940 |
| B4.16 | 72.54 | 98.505 | 0.687 | 0.942 |
| B4.17 | 72.84 | 95.338 | 0.703 | 0.941 |
| B4.18 | 72.99 | 95.838 | 0.692 | 0.942 |
| B4.19 | 72.52 | 97.525 | 0.725 | 0.941 |

Table 30 above contains total statistics for all the items in B4 for research objective 3. Please see Appendix A on page 32 for the item- statistics for the research objectives 1, 2 and 3 and the correlation matrix for research objective 3–B4.

### 4.2.2. Reliability Statistics of Empirical Factors

To confirm reliability and internal consistency, the reliability of the empirical factors was identified, and Table 31 below depicts the reliability statistics for research objective 1–B2–factor 1.

**Table 31.** Reliability statistics for research objective 1–B2–factor 1.

| Reliability Statistics | |
| --- | --- |
| **Cronbach's Alpha** | **N of Items** |
| 0.880 | 6 |

As seen in Table 31 above, Cronbach's alpha is 0.880, which is greater than 0.7; therefore, it is reliable. Table 32 below shows the reliability statistics for research objective 1–B2–factor 2.

**Table 32.** Reliability statistics for research objective 1–B2–factor 2.

| Reliability Statistics | |
| --- | --- |
| **Cronbach's Alpha** | **N of Items** |
| 0.861 | 7 |

Table 32 above shows a Cronbach alpha value of 0.861, which is greater than 0.7; therefore, it is reliable. Table 33 below illustrates the reliability statistics for research objective 2–B3–factor 1.

**Table 33.** Reliability statistics for research objective 2–B3–factor 1.

| Reliability Statistics | |
| --- | --- |
| **Cronbach's Alpha** | **N of Items** |
| 0.941 | 12 |

As seen in Table 33 above, the Cronbach alpha is 0.941, which is above 0.7; so, it is reliable. Table 34 below illustrates the reliability statistics for research objective 2–B3–factor 2.

**Table 34.** Reliability statistics for research objective 2–B3–factor 2.

| Reliability Statistics | |
| --- | --- |
| **Cronbach's Alpha** | **N of Items** |
| 0.903 | 5 |

Table 34 above has a Cronbach alpha value of 0.903, which was above 0.7; therefore, it is reliable. The reliability statistics for research objective 2–B3–factor 3 is illustrated in Table 35 below.

**Table 35.** Reliability statistics for research objective 2–B3–factor 3.

| Reliability Statistics | |
| --- | --- |
| **Cronbach's Alpha** | **N of Items** |
| 0.895 | 5 |

As shown in Table 35 above, the Cronbach alpha is 0.895, which is greater than 0.7; therefore, it is reliable. Table 36 below shows the reliability statistics for research objective 2–B3–factor 4.

**Table 36.** Reliability statistics for research objective 2–B3–factor 4.

| Reliability Statistics | |
| --- | --- |
| **Cronbach's Alpha** | **N of Items** |
| 0.826 | 2 |

As shown in Table 36 above, the Cronbach Alpha is 0.826, which is greater than 0.7; therefore, it is reliable. Table 37 below shows the reliability statistics for research objective 3–B4–factor 1.

**Table 37.** Reliability statistics for research objective 3–B4–factor 1.

| Reliability Statistics | |
| --- | --- |
| **Cronbach's Alpha** | **N of Items** |
| 0.926 | 9 |

As shown in Table 37 above, the Cronbach alpha is 0.926, so it is reliable. Table 38 below shows the reliability statistics for research objective 3–B4–factor 2.

**Table 38.** Reliability statistics for research objective 3–B4–factor 2.

| Reliability Statistics | |
| --- | --- |
| **Cronbach's Alpha** | **N of Items** |
| 0.881 | 6 |

Table 38 above shows a Cronbach alpha of 0.881; therefore, it is reliable. Table 39 below shows the reliability statistics for research objective 3–B4–factor 3.

**Table 39.** Reliability statistics for research objective 3–B4–factor 3.

| Reliability Statistics | |
| --- | --- |
| **Cronbach's Alpha** | **N of Items** |
| 0.836 | 4 |

As shown in Table 39 above, the Cronbach alpha is 0.836, which is higher than 0.7, and this confirms its reliability.

## 5. Results and Discussion

The respondents were asked to answer questions on work culture related to interface risks. As depicted in Table 7 above, 1 respondent strongly disagreed that interface risks between project stakeholders can be classified as uncertainties, which represents 0.5% of the total responses; 15 (7.3%) respondents disagreed; 39 (19.0%) respondents were neutral; 129 (62.9%) agreed; while 21 (10.2%) of the respondents strongly agreed. A total of 1 respondent strongly disagreed that interface risks between project stakeholders can be classified as unidentified risks, representing 0.5% of the responses; 12 (5.9%) disagreed with the statement; 57 (27.8%) respondents were neutral; 118 (57.6%) agreed; while 17 (8.3%) respondents strongly agreed with the statement. The responses showed that 119 (58%) respondents agreed that the identification of hard interface risks encourages effective collaboration between project stakeholders, while 72 (35.1%) respondents strongly agreed. A total of 107

(52.2%) respondents agreed that the identification of soft interface risks encourages effective collaboration between project stakeholders, while 65 (31.7%) respondents strongly agreed.

For research objective 1, Spearman's rho showed that there is a correlation between the consequences of poor and ineffective interface risk management approaches and their influence on a project, since the values of the Spearman's coefficient are greater than 0.3 and, as shown in Table 8 above, for Bartlett's test of sphericity, the significance, i.e., the *p* value, was less than 0.001, which was less than 0.05. This means that the higher the probability of the consequences, such as project delays, poor quality, industrial actions, additional costs, etc., the higher the impacts on the project. The Kaiser–Meyer–Olkin measure of sampling adequacy was 0.898, which was greater than 0.6 and demonstrates that factor analysis can be carried out.

For the research objective 2, Spearman's rho showed that there was a correlation between the interface risk management approaches and their influences on the project goals and objectives and the successful execution of construction projects, since the values of Spearman's coefficient were greater than 0.3 and, as shown in Table 9 above, for Bartlett's test of sphericity, the significance, i.e., the *p* value, was 0.000, which was less than 0.05. This means that the higher the probability of the interface risk management approaches, such as defining standard methods and procedures, creating a virtual construction model during the construction phase, establishing a building information modelling (BIM) volume strategy, etc., the higher the impacts on the project goals and objectives. The Kaiser–Meyer–Olkin measure of sampling adequacy was 0.915, which was greater than 0.6, which shows that factor analysis can be carried out.

For the research objective 3, Spearman's rho showed that there was correlation between the extent to which the following causes of interface risks influences construction projects, since the values of Spearman's coefficient are greater than 0.3. And, as shown in Table 10 above, for the Bartlett's test of sphericity, the significance, *p* value was 0.000, which was less than 0.05. This means that the higher the probability of the causes of interface risks, such as incompetency, poor inventories, lack of knowledge sharing, procurement delays, etc., the higher the impacts on the project execution. The Kaiser–Meyer–Olkin measure of sampling adequacy was 0.917, which was greater than 0.6 and shows that factor analysis can be carried out.

*Proposed New Framework*

Discussion

This study investigated interface risks, the various causes of interface risks, the consequences of poor interface risk management and their levels of influence on projects, and interface risk management approaches by organisations and how they influence the overall project objectives. Figure 4 below depicts the new proposed framework to identify and manage interface risks in construction projects.

As shown in Figure 4 above, for an effective interface risk assessment, it is recommended that construction industry must effectively establish a building information modelling volume strategy and create a virtual construction model during the construction phase; in addition, construction supply chain risks must be carefully identified during the interfaces establishment stages; interface risks must be carefully identified during the conceptualisation; and planning, construction, and execution stages and standard methods and procedures must be defined to effectively identify and manage interface risks as the occur in the project lifecycle. Effective stakeholder management is also crucial for effective interface risk management since many interface risks are created by the numerous stakeholders involved in the project. The proposed stakeholder management approaches will be shown in Figure 5 below.

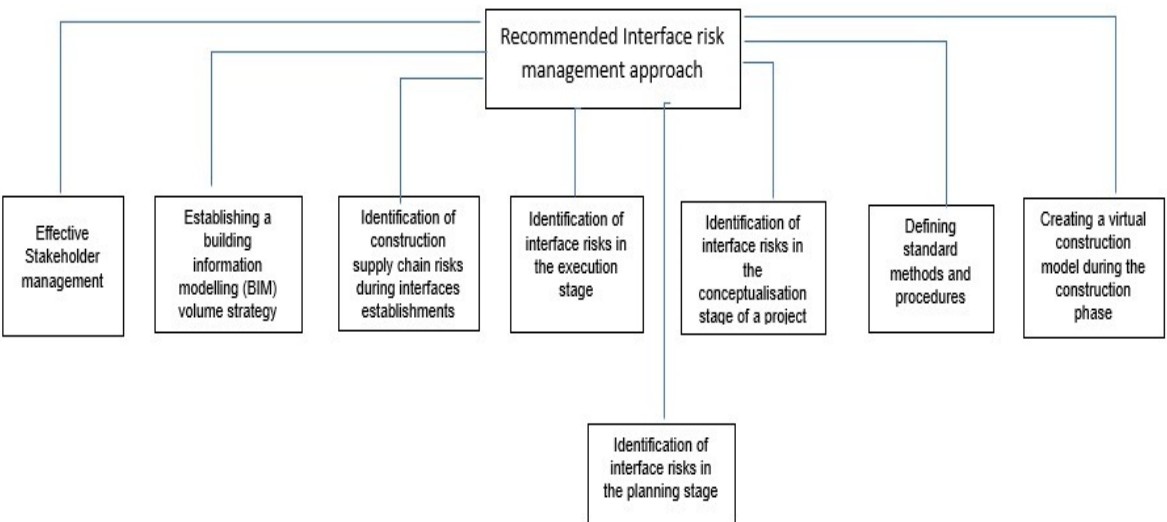

**Figure 4.** Proposed interface risk assessment approach framework.

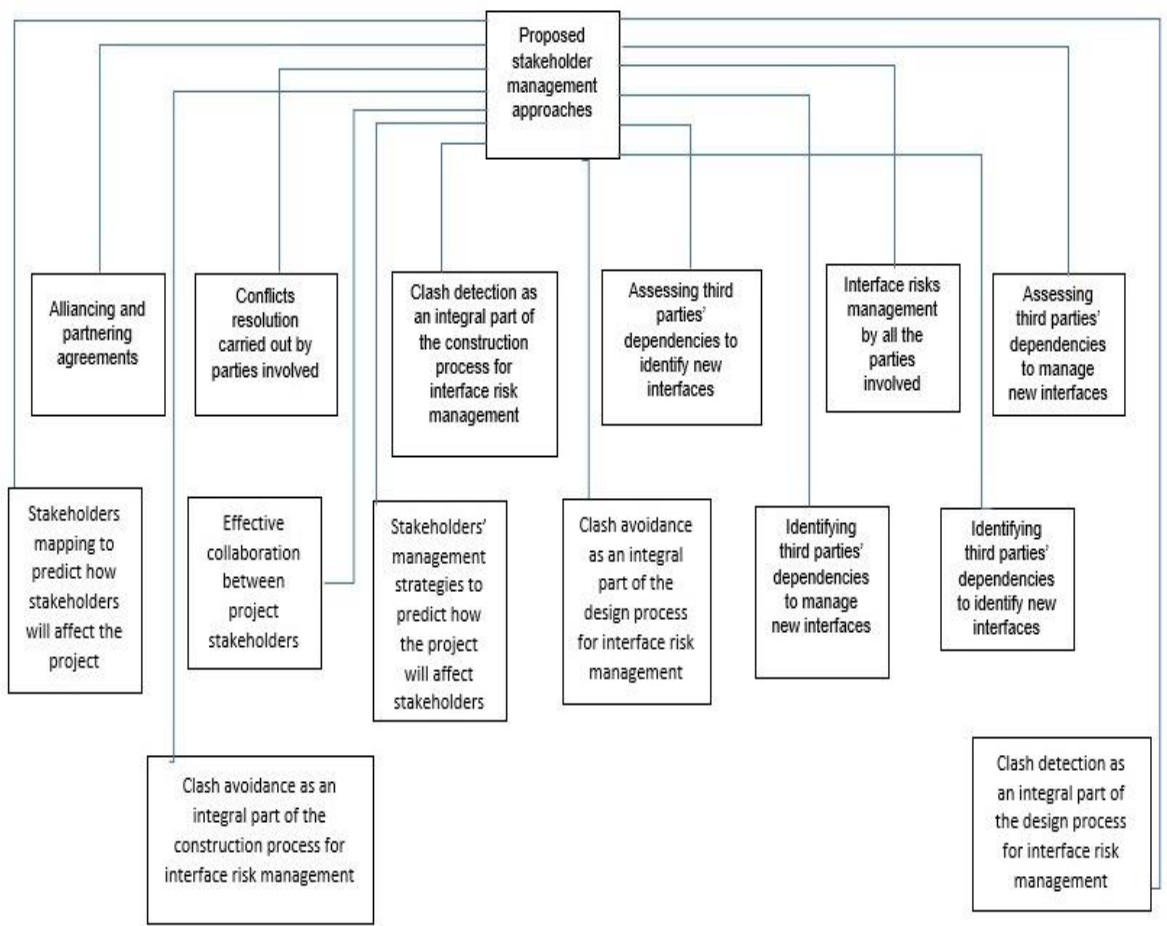

**Figure 5.** Proposed stakeholder management approach framework.

As shown in the proposed framework in Figure 5, for an effective stakeholder management in construction projects, it is recommended that construction industry must encourage and implement alliancing and partnering agreements, conflicts resolution must be carried out by every party involved in the project, clash detection and avoidance must integrated into the construction and design processes for interface risk management, third parties' dependencies must be carefully and continually identified and assessed in order to identify

and manage new interfaces, and interface risk management must be carried out by all parties involved for overall project success. Stakeholder mapping must be carried out to determine how stakeholders influence the project and how the project influences the stakeholders. Figure 6 below shows the proposed interface risk mitigation methods.

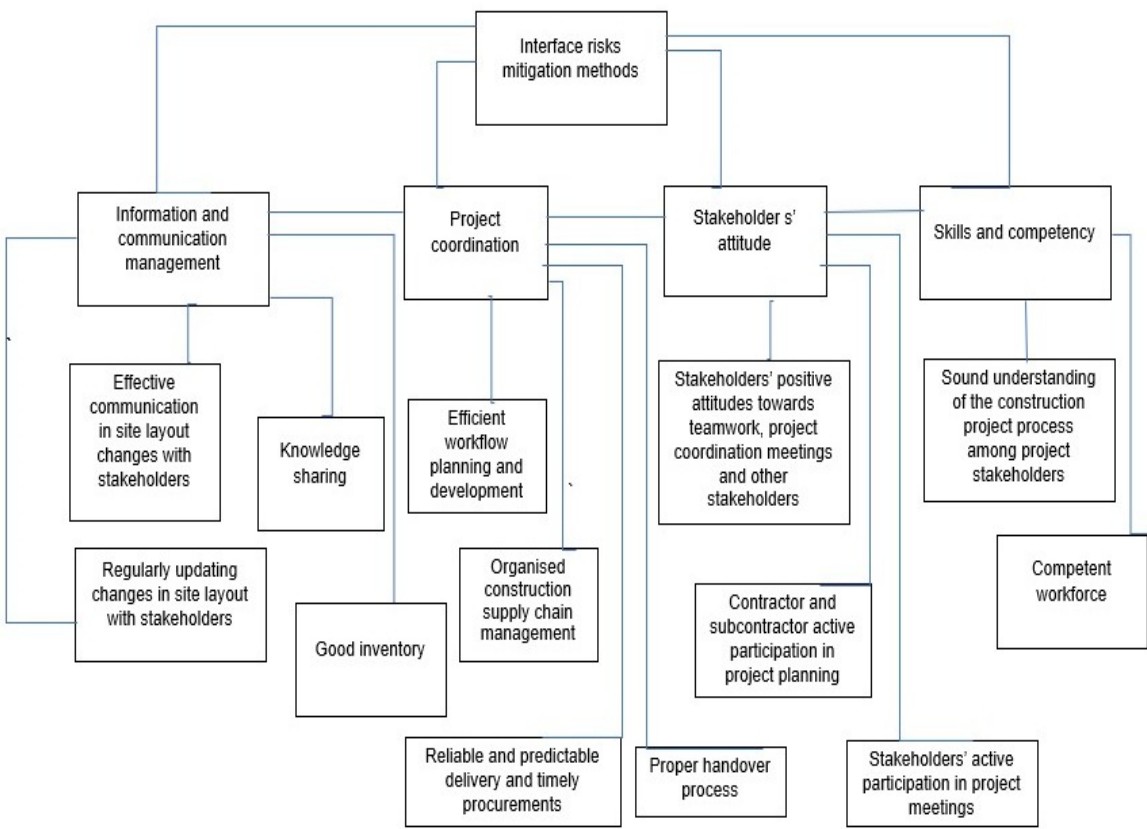

**Figure 6.** Proposed interface risk mitigation methods.

The proposed framework in Figure 6 above ensures that information and communication are effectively managed throughout the project, that project coordination is improved and effective, that stakeholders' attitudes produce positive outcomes, and that appropriate skills and competent workers are adequately utilized in all phases of the project to minimise interface risk occurrence and effectively manage them as they occur in the project in order to realise project objectives. The last chapter will focus on the conclusion of the results, findings, proposed frameworks and recommendations.

## 6. Conclusions

Interface risk is one of the major challenges facing the construction industry because construction projects are complex by nature—involving many activities and participants with different responsibilities and tasks. It is crucial to carefully identify and manage these risks arising from the interfaces since they are inherent in all the construction project phases according to the findings of this research. This study revealed that most construction projects encountered interface risks throughout a project lifecycle, and if they were not carefully and properly identified and managed during the project, they had negative influences on project objectives and could evidently lead to project failure or abandonment. Interface risks must be continually identified and managed during the conceptualisation, planning, and interface establishment phases and must be carefully assessed, monitored, and managed throughout the project. Stakeholder management was identified as one of the main ways to effectively manage interface risks because stakeholders play vital roles in interface risk management.

The effective communication, knowledge, and information sharing among project stakeholders have positive impacts on the success of the project because coordinated and evenly distributed knowledge and information flow on site layout changes and construction processes facilitate successful project delivery as well as identifying both soft and hard interface risks, as these encourage effective collaboration, alliancing, and partnering agreements between project stakeholders and, in essence, mitigate conflicts and clashes among stakeholders. The integration of clash detection and avoidance into a project's lifecycle is vital for success, as this provides the project team with initial tools and ideas to effectively identify and manage such events and mitigate them in advance. Suppliers, vendors, clients, contractors, and subcontractors must communicate effectively to ensure timely and reliable delivery, procurement, and timely funding in order not to delay the project or incur additional costs. Clients must provide timely and sufficient funding because poor and delayed funding results in supply chain disruptions and labour and material delays, which ultimately prolong the project and result in additional costs. Effective interface risk management in construction projects will minimise and save costs and time; mitigate industrial action and damage claim; improve and maintain project quality and safety; protect the environment; facilitate good workflow planning and development; and protect the reputation of the organisation that would have been damaged as a result of regulatory infringements, industrial actions, damage claims, extended projected delivery time, stakeholder complaints, project abandonment, and failure. Identifying and assessing parties' dependencies to identify and manage new interfaces is vital for project success. For effective interface risk management, standard methods and procedures must be defined, a building information modelling volume strategy must be established and utilised effectively, and a virtual construction model must be created and monitored regularly. Regular meetings with stakeholders facilitate effective interface risk management because every stakeholder will have updated and firsthand information on the project's progress and schedule. Stakeholders' attitudes towards project coordination are vital to project success as they determine how effectively each project phase will be completed. Clash detection and avoidance must be integrated into the planning, design, and construction stages and conflicts must be resolved by every party involved. Effective construction supply chain management is important in project delivery, and procurement deliveries must be timely, predictable, and reliable to avoid material and labour shortages or surplus, and inventories must be updated regularly for effective project site coordination and workflows. An incompetent labour force; a poor understanding of construction project processes; and contractors', clients', and subcontractors' negative attitudes generate many interface risks, and these must be carefully identified and managed during the planning and contracting stages of a project. Changes in site layout must be updated and communicated to project participants. To save time; minimise costs; and maintain anticipated project quality, safety, and standards, interface risks must be carefully identified and managed by project participants, and every stakeholder must participate in project coordination meetings, comply with the project guidelines, and actively participate in identifying and managing interface risks throughout the project for its successful execution. Implementing the proposed stakeholder management framework and integrating it into the interface risk management framework will greatly minimise the risks involved that are generated by every party in a construction project.

**Author Contributions:** Conceptualization, M.C.O., A.V. and J.-H.C.P.; investigation, M.C.O.; methodology, M.C.O.; supervision, A.V. and J.-H.C.P.; writing—original draft, M.C.O.; writing—review and editing, A.V. and J.-H.C.P. All authors have read and agreed to the published version of the manuscript.

**Funding:** This research is funded by the University of Johannesburg, South Africa.

**Data Availability Statement:** All data and materials are available on request from the corresponding author.

**Conflicts of Interest:** The authors declare no conflicts of interest.

## Appendix A

Table A1 above shows the mean scores and standard deviations of all the Section B2 items—research objective 1 ("The consequences of poor and ineffective interface risk management approach."). The mean scores were in between "moderate extent" (3) and slightly above "large extent" (4), which means for each item, most respondents chose in between "moderate extent" (slightly above "moderate extent") and slightly towards "very large extent"—a bit above "large extent".

**Table A1.** Item statistics for research objective one.

| | Item Statistics | | |
|---|---|---|---|
| | **Mean** | **Std. Deviation** | **N** |
| B2.1 | 4.00 | 0.789 | 205 |
| B2.2 | 4.03 | 0.813 | 205 |
| B2.3 | 4.35 | 0.723 | 205 |
| B2.4 | 4.06 | 0.826 | 205 |
| B2.5 | 3.81 | 0.845 | 205 |
| B2.6 | 4.22 | 0.751 | 205 |
| B2.7 | 3.73 | 0.805 | 205 |
| B2.8 | 4.13 | 0.756 | 205 |
| B2.9 | 4.15 | 0.809 | 205 |
| B2.10 | 3.91 | 0.729 | 205 |
| B2.11 | 4.24 | 0.734 | 205 |
| B2.12 | 4.00 | 0.741 | 205 |
| B2.13 | 4.17 | 0.744 | 205 |

Table A2 above shows the mean scores and standard deviations of all the items in Section B3—research objective 2 ("Interface risk management approaches by organisation."). The mean scores were in between "moderate extent" (3) and slightly above "large extent" (4), which means for each item, most respondents chose in between "moderate extent" and slightly towards "very large extent".

**Table A2.** Item statistics for research objective two.

| | Item Statistics | | |
|---|---|---|---|
| | **Mean** | **Std. Deviation** | **N** |
| B3.1 | 4.22 | 0.697 | 205 |
| B3.2 | 3.91 | 0.765 | 205 |
| B3.3 | 3.91 | 0.861 | 205 |
| B3.4 | 3.85 | 0.785 | 205 |
| B3.5 | 3.94 | 0.884 | 205 |
| B3.6 | 4.14 | 0.782 | 205 |
| B3.7 | 4.22 | 0.779 | 205 |
| B3.8 | 4.10 | 0.834 | 205 |
| B3.9 | 4.12 | 0.792 | 205 |
| B3.10 | 4.16 | 0.704 | 205 |
| B3.11 | 4.23 | 0.774 | 205 |
| B3.12 | 4.25 | 0.687 | 205 |
| B3.13 | 4.08 | 0.813 | 205 |
| B3.14 | 4.12 | 0.700 | 205 |
| B3.15 | 4.08 | 0.819 | 205 |
| B3.16 | 3.94 | 0.683 | 205 |
| B3.17 | 3.91 | 0.853 | 205 |
| B3.18 | 3.92 | 0.723 | 205 |
| B3.19 | 3.84 | 0.860 | 205 |
| B3.20 | 3.88 | 0.747 | 205 |
| B3.21 | 3.93 | 0.808 | 205 |
| B3.22 | 4.18 | 0.694 | 205 |
| B3.23 | 4.34 | 0.766 | 205 |
| B3.24 | 4.30 | 0.689 | 205 |

The above Table A3 contains the mean and standard deviations of Section B4.

The mean scores were in between "moderate extent" (3) and slightly above "large extent" (4), which means for each item, most respondents chose in between "moderate extent" and slightly towards "very large extent".

**Table A3.** Item statistics for research objective three.

| | Mean | Std. Deviation | N |
|---|---|---|---|
| | **Item Statistics** | | |
| B4.1 | 4.29 | 0.680 | 205 |
| B4.2 | 4.00 | 0.728 | 205 |
| B4.3 | 4.02 | 0.770 | 205 |
| B4.4 | 3.97 | 0.754 | 205 |
| B4.5 | 4.00 | 0.813 | 205 |
| B4.6 | 4.25 | 0.742 | 205 |
| B4.7 | 4.22 | 0.740 | 205 |
| B4.8 | 4.01 | 0.764 | 205 |
| B4.9 | 4.35 | 0.750 | 205 |
| B4.10 | 4.24 | 0.727 | 205 |
| B4.11 | 3.94 | 0.768 | 205 |
| B4.12 | 3.82 | 0.827 | 205 |
| B4.13 | 4.04 | 0.827 | 205 |
| B4.14 | 4.25 | 0.701 | 205 |
| B4.15 | 4.03 | 0.785 | 205 |
| B4.16 | 3.94 | 0.725 | 205 |
| B4.17 | 3.63 | 0.928 | 205 |
| B4.18 | 3.49 | 0.905 | 205 |
| B4.19 | 3.96 | 0.756 | 205 |

As shown in Table A4 above, many of the correlations were greater than 0.3. B4.2 and B4.1 had a correlation of 0.267, B4.4 and B4.13 had a correlation of 0.277, B4.6 and B4.17 had a correlation of 0.275, B4.2 and B4.17 had a correlation of 0.276, B4.11 and B4.14 had a correlation of 0.266, B4.2 and B4.9 had a correlation of 0.261, B4.17 and B4.14 had a correlation of 0.231, and B4.13 and B4.14 had a correlation of 0.287.

**Table A4.** Correlation matrix for research objective 3–B4.

| | | B4.1 | B4.2 | B4.3 | B4.4 | B4.5 | B4.6 | B4.7 | B4.8 | B4.9 | B4.10 | B4.11 | B4.12 | B4.13 | B4.14 | B4.15 | B4.16 | B4.17 | B4.18 | B4.19 |
|---|---|---|---|---|---|---|---|---|---|---|---|---|---|---|---|---|---|---|---|---|
| | | | | | | | | | **Correlation Matrix** | | | | | | | | | | | |
| Correlation | B4.1 | 1.000 | 0.267 | 0.539 | 0.437 | 0.600 | 0.331 | 0.648 | 0.360 | 0.557 | 0.390 | 0.561 | 0.362 | 0.467 | 0.340 | 0.543 | 0.373 | 0.504 | 0.417 | 0.549 |
| | B4.2 | 0.267 | 1.000 | 0.324 | 0.563 | 0.381 | 0.536 | 0.337 | 0.459 | 0.261 | 0.500 | 0.325 | 0.570 | 0.415 | 0.452 | 0.360 | 0.539 | 0.276 | 0.461 | 0.419 |
| | B4.3 | 0.539 | 0.324 | 1.000 | 0.517 | 0.665 | 0.341 | 0.636 | 0.366 | 0.503 | 0.375 | 0.567 | 0.361 | 0.445 | 0.370 | 0.575 | 0.354 | 0.589 | 0.461 | 0.583 |
| | B4.4 | 0.437 | 0.563 | 0.517 | 1.000 | 0.456 | 0.522 | 0.451 | 0.631 | 0.409 | 0.604 | 0.404 | 0.526 | 0.277 | 0.468 | 0.374 | 0.436 | 0.426 | 0.452 | 0.368 |
| | B4.5 | 0.600 | 0.381 | 0.665 | 0.456 | 1.000 | 0.363 | 0.609 | 0.410 | 0.536 | 0.421 | 0.597 | 0.446 | 0.502 | 0.307 | 0.583 | 0.424 | 0.619 | 0.509 | 0.534 |
| | B4.6 | 0.331 | 0.536 | 0.341 | 0.522 | 0.363 | 1.000 | 0.451 | 0.530 | 0.503 | 0.669 | 0.355 | 0.511 | 0.327 | 0.559 | 0.374 | 0.455 | 0.275 | 0.437 | 0.317 |
| | B4.7 | 0.648 | 0.337 | 0.636 | 0.451 | 0.609 | 0.451 | 1.000 | 0.480 | 0.679 | 0.435 | 0.621 | 0.417 | 0.498 | 0.383 | 0.613 | 0.372 | 0.592 | 0.493 | 0.570 |
| | B4.8 | 0.360 | 0.459 | 0.366 | 0.631 | 0.410 | 0.530 | 0.480 | 1.000 | 0.428 | 0.647 | 0.470 | 0.547 | 0.240 | 0.533 | 0.424 | 0.533 | 0.444 | 0.436 | 0.358 |
| | B4.9 | 0.557 | 0.261 | 0.503 | 0.409 | 0.536 | 0.503 | 0.679 | 0.428 | 1.000 | 0.508 | 0.541 | 0.353 | 0.460 | 0.411 | 0.590 | 0.399 | 0.439 | 0.336 | 0.503 |
| | B4.10 | 0.390 | 0.500 | 0.375 | 0.604 | 0.421 | 0.669 | 0.435 | 0.647 | 0.508 | 1.000 | 0.379 | 0.536 | 0.253 | 0.611 | 0.417 | 0.539 | 0.315 | 0.457 | 0.394 |
| | B4.11 | 0.561 | 0.325 | 0.567 | 0.404 | 0.597 | 0.355 | 0.621 | 0.470 | 0.541 | 0.379 | 1.000 | 0.546 | 0.575 | 0.266 | 0.670 | 0.434 | 0.635 | 0.440 | 0.553 |
| | B4.12 | 0.362 | 0.570 | 0.361 | 0.526 | 0.446 | 0.511 | 0.417 | 0.547 | 0.353 | 0.536 | 0.546 | 1.000 | 0.590 | 0.473 | 0.536 | 0.677 | 0.491 | 0.626 | 0.489 |
| | B4.13 | 0.467 | 0.415 | 0.445 | 0.277 | 0.502 | 0.327 | 0.498 | 0.240 | 0.460 | 0.253 | 0.575 | 0.590 | 1.000 | 0.287 | 0.594 | 0.437 | 0.523 | 0.465 | 0.637 |
| | B4.14 | 0.340 | 0.452 | 0.370 | 0.468 | 0.307 | 0.559 | 0.383 | 0.533 | 0.411 | 0.611 | 0.266 | 0.473 | 0.287 | 1.000 | 0.441 | 0.597 | 0.231 | 0.370 | 0.428 |
| | B4.15 | 0.543 | 0.360 | 0.575 | 0.374 | 0.583 | 0.374 | 0.613 | 0.424 | 0.590 | 0.417 | 0.670 | 0.536 | 0.594 | 0.441 | 1.000 | 0.605 | 0.647 | 0.490 | 0.696 |
| | B4.16 | 0.373 | 0.539 | 0.354 | 0.436 | 0.424 | 0.455 | 0.372 | 0.533 | 0.399 | 0.539 | 0.434 | 0.677 | 0.437 | 0.597 | 0.605 | 1.000 | 0.456 | 0.597 | 0.541 |
| | B4.17 | 0.504 | 0.276 | 0.589 | 0.426 | 0.619 | 0.275 | 0.592 | 0.444 | 0.439 | 0.315 | 0.635 | 0.491 | 0.523 | 0.231 | 0.647 | 0.456 | 1.000 | 0.741 | 0.662 |
| | B4.18 | 0.417 | 0.461 | 0.461 | 0.452 | 0.509 | 0.437 | 0.493 | 0.436 | 0.336 | 0.457 | 0.440 | 0.626 | 0.465 | 0.370 | 0.490 | 0.597 | 0.741 | 1.000 | 0.576 |
| | B4.19 | 0.549 | 0.419 | 0.583 | 0.368 | 0.534 | 0.317 | 0.570 | 0.358 | 0.503 | 0.394 | 0.553 | 0.489 | 0.637 | 0.428 | 0.696 | 0.541 | 0.662 | 0.576 | 1.000 |

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
