# Peer review of "A Systematic Approach to Identify and Manage Interface Risks between Project Stakeholders in Construction Projects"

_2673-4109, doi:10.3390/civileng5010005_

Round 1
Reviewer 1 Report
Comments and Suggestions for Authors
This paper explores the relationships between project stakeholders and how the interface risks between them that influence project execution can be identified and managed for the overall construction project success. The topic is interesting. Here are my comments to improve the paper:
1- your abstract reads well overall. Just one suggestion: add 1-2 sentences presenting a brief summary of your interpretations and conclusions.
2- In the introduction, make clearer what knowledge gaps you identified and how your research addresses them. Also, make the research objectives/questions clearer. Answer the “so what?” question. Why investigating such matter is important? End the introduction with an outline of the paper; what comes next?
3- The novelty/originality should be clearly justified that the manuscript contains sufficient contributions to the new body of knowledge from the international perspective. What new things (new theories, new methods, or new policies) can the paper contribute to the existing international literature? This point must be reasonably justified by a Literature Review, clearly introduced in Introduction Section, and completely discussed in Discussion Section.
4- your study discusses costs in a number of places, but it ignores transaction costs – or at least doesn’t explicitly discuss transaction costs. It would be helpful if you could acknowledge these hidden costs somewhere in your paper. Here are some relevant references:
https://doi.org/10.1080/15623599.2020.1738204
https://doi.org/10.1177/08854122211062085
https://doi.org/10.1108/K-10-2014-0206
5- it is a bit weird to have a section on research objectives in page 4. They should move to the introduction section where you are discussing your paper and its contributions.
6- your methodology section is super short and doesn’t provide the reader with any details of the methodological steps and approaches. In the revised version, I would like to see a comprehensive methodology section with as much as details possible so other researchers can replicate your work if needed.
7- What are the limitations of your study/methodology?
8- you have included way too many tables. Go through the paper and check whether they all need to be included or can be removed / moved to the annex. At the moment, it is hard to follow the paper as it gets interrupted by so many tables. Do you expect people to read your paper? Would you read a paper like this yourself?
9- you need to further elaborate on your conclusions. What do your results mean for different stakeholders.
10- The conclusion could do more to tease out the wider resonance of the paper for the journal's international readership.
Comments on the Quality of English Language
Minor editing of English language required
Author Response
Reviewer report 1
- your abstract reads well overall. Just one suggestion: add 1-2 sentences presenting a brief summary of your interpretations and conclusions.
My response:
Effective stakeholder management is crucial for effective interface risks management since a lot of interface risks are created by the numerous stakeholders involved in the project and the proposed frameworks will effectively mitigate the consequences and causes of interface risks and to effectively mitigate these risks, it involve effective stakeholder management, building information modelling volume strategy, creating a virtual construction model during the construction phase, construction supply chain risks must be carefully identified during the interfaces establishment stages, interface risks must be carefully identified during the conceptualisation, planning, construction and execution stages and standard methods and procedures must be defined to effectively identify and manage interface risks as the occur in the project lifecycle plus implementing the proposed risk mitigation frameworks.
- In the introduction, make clearer what knowledge gaps you identified and how your research addresses them. Also, make the research objectives/questions clearer. Answer the “so what?” question. Why investigating such matter is important? End the introduction with an outline of the paper; what comes next?
My response:
.A formal risk management plan is barely used in executing construction projects because contractors and subcontractors rely on their past experiences and judgements and these result in unforeseen circumstances that can negatively impact project objectives since they are not fully equipped with the tools to effectively manage the unidentified risks and uncertainties associated with interfaces and numerous stakeholders. Stakeholder management is not effectively incorporated in the risk management plan as numerous stakeholders and their roles and influences are not carefully identified in the conceptualisation stage of the project and this give rise to additional interface risks.
Research Objective
The study objective was to carry out literature review on interfaces in construction, interface risks and interface risks management. The study three main objectives are:
- To identify the consequences of poor and ineffective interface risks management approach and how they influenced construction project delivery
- To identify the current Interface risks management methods utilised by organisations
- To identify the causes of interface risks and how they influenced project objectives
To support the objectives of the study, these five research questions were asked. Respondents were asked to identify:
- The consequences of poor and ineffective interface risks management approach
- The interface risks management approaches implemented by their organisations
- The causes of interface risks
- The novelty/originality should be clearly justified that the manuscript contains sufficient contributions to the new body of knowledge from the international perspective. What new things (new theories, new methods, or new policies) can the paper contribute to the existing international literature? This point must be reasonably justified by a Literature Review, clearly introduced in Introduction Section, and completely discussed in Discussion Section
My response:
Stakeholder management is one of the major challenges faced in construction projects as these numerous stakeholders have personal and diverse interests and objectives ranging from personal monetary gains, political and economic interests, opportunities and favours and these in essence compromise the objectives of the project to successfully and timely complete the project. The paper identified and proposed new methods to manage stakeholders and in essence mitigate interface risks form these stakeholders and interfaces.
- your study discusses costs in a number of places, but it ignores transaction costs – or at least doesn’t explicitly discuss transaction costs. It would be helpful if you could acknowledge these hidden costs somewhere in your paper. Here are some relevant references:
My response:
Transaction cost is one of the contributing factors to interface risks and according to Standish Group released reports, project costs are routinely surpassed by 21% to 100% (60% of project cost overruns). Furthermore, 4% of initiatives had a cost overrun that cause the project's final cost to be higher than four times previously predicted. Standish believed group analysis of construction projects in the United Kingdom revealed cost variances ranging from 50 to 80%, which is significant for a construction project [22, 23]. According to [23], some of the factors that affect transaction costs are payment schedule, quality of project, type of project, value of project, complexity of the project, similar experience, parties’ relationships, change order, efficiency of the organisation, duration of the project and many more and if these factors are not carefully considered during the project lifecycle, the projects will be adversely affected. Some of these costs are incurred to secure the delivery of high-quality developments or environmental enhancement, but it is often evident that the costs are exorbitant, and a procedure is overly long and inefficient. Identifying and decreasing such costs assists planners and decision- makers in improving the efficiency, efficacy, and acceptability of their projects and processes [24, 25, 26].
- it is a bit weird to have a section on research objectives in page 4. They should move to the introduction section where you are discussing your paper and its contributions.
My response:
The study focuses on a systematic approach in identifying and managing risks associated with every interface in construction projects in every phase. Literature review was done to identify critical areas of knowledge of the field of study, with the purpose of presenting a summary of recent literature on the topic. The primary objective of the study is to develop a framework on how to identify and manage interface risks in construction for overall project success.
The paper was outlined in this order: study background was discussed after the introduction then chapter three discussed the materials and methods used, chapter four was the findings and analysis, results and discussions were discussed in chapter five followed by conclusion in chapter six.
- your methodology section is super short and doesn’t provide the reader with any details of the methodological steps and approaches. In the revised version, I would like to see a comprehensive methodology section with as much as details possible so other researchers can replicate your work if needed.
My response:
These three Likert-type scale response anchors were chosen for the questionnaire in order to find out the level of agreement with the individual statements in the questionnaire, the frequencies of each statement or items in the questionnaire and the extent scale was used to find out the extent in which each statement or item in the questionnaire influences construction projects. The data collection process commenced by administering a biographical questionnaire to ascertain the appropriate research participants in section A, this includes size of organisation, profession, age and highest academic qualifications. Section B had three subsections namely B2, B3, and B4. Section B2 involved questions related to the consequences of poor and ineffective interface risks management approach, where respondents were asked to identify these consequences and rate them according to the extent scale. Section B3 are questions related to interface risks management methods currently adopted by their organisations and the extent they influenced project objectives and section B4 were questions related to the causes of interface risks and the respondents were asked to identify the extents the causes of interface risks influenced the successful execution of construction projects.
The items or questions in each section were coded for ease of analysis. For the five-point Linkert scale chosen, rarely was coded as a 2, sometimes coded as a 3, often was coded as a 4 and always was coded as a 5. Also, strongly disagree was coded as a 1, disagree was coded as a 2, neutral was coded as a 3, agree was coded as a 4 and strongly agree was coded as a 5 then, to no extent was coded as a 1, small extent was coded as a 2, moderate extent was coded as a 3, large extent was coded as a 4 and very large extent was coded as a 5.
- What are the limitations of your study/methodology?
My response:
The study focused only on the South African construction industry and active industry professionals. All respondents were South African residents and had experience in South African construction industry. Industry professionals living outside of South Africa were not part of the study. Every participants was either involved in the conceptualisation, planning, contracting, subcontracting, procurement, construction, execution, HSE management and commissioning phases of the projects.
- you have included way too many tables. Go through the paper and check whether they all need to be included or can be removed / moved to the annex. At the moment, it is hard to follow the paper as it gets interrupted by so many tables. Do you expect people to read your paper? Would you read a paper like this yourself?
My response:
Although, the tables were much but I chose to include them on each chapter so that readers will fully understand how the tables were generated and explained to maintain a continual information flow without referring them to appendix for reference.
- you need to further elaborate on your conclusions. What do your results mean for different stakeholders.
My response:
Stakeholder management was identified as one of the major ways to effectively manage interface risks because stakeholders play vital roles in interface risk management.
Effective communication, knowledge and information sharing among project stakeholders have positive impacts on the success of the project because coordinated and evenly distributed knowledge and information flow on site layouts changes and construction processes facilitate successful project delivery as well as identifying both soft and hard interface risks as these encourage effective collaboration, alliancing and partnering agreements between project stakeholders, mitigate conflicts and clashes among stakeholders. Clash detection and avoidance integrated into the project lifecycle in every project are vital its success as these will provide the project team initial tools and idea to effectively identify and manage such events and mitigate them before-hand. Suppliers, vendors, clients, contractors and subcontractors must communicate effectively to ensure timely and reliable delivery, procurement and timely funding in order not to delay the project and incur additional costs. Clients must provide timely and sufficient funding because poor and delayed funding results in supply chain disruptions, labour and materials delays which eventually prolong the project and result in additional costs. Effective interface risks management in construction projects will minimise and save cost.
Effective construction supply chain management is important in project delivery and procurement deliveries must be timely, predictable and reliable to avoid materials and labour shortages or surplus and inventories must be updated regularly for effective project site coordination and workflows.
- The conclusion could do more to tease out the wider resonance of the paper for the journal's international readership.
My response:
Regular meetings with stakeholders facilitate effective interface risks management because every stakeholder will have updated and firsthand information on project progress and schedule. Stakeholders attitude towards project coordination is vital to project success. Clash detection and avoidance must be integrated in the planning, design and construction stages and conflicts must be resolved by every party involved. Effective construction supply chain management is important in project delivery and procurement deliveries must be timely, predictable and reliable to avoid materials and labour shortages or surplus and inventories must be updated regularly for effective project site coordination and workflows.
Implementing the proposed stakeholder management framework and integrating it into the interface risk management frameworks will greatly minimise the risks involved and generated by every party in a construction project.
Thank you for your recommendations. I learnt a lot from you and your suggestions will be useful to me henceforth.
The changes were added to the paper.
Please see the attachment.

Reviewer 2 Report
Comments and Suggestions for Authors
The manuscript is to provide a systematic approach on identifying and managing interface risks between project stakeholders in construction projects. The construction industry faces interface risks which are the most encountered problem in the construction industry. THe research objective illustrates three major objectives in this manuscript without identifying the interface risks. We need to first know what interface risks are, before we try to understand how to manage interface risks and understand what causes of interface risks are. More comments and suggestions are given below.
(1) As the title of the manuscript, one needs to identify interface risks between project stakeholders in construction projects. The interface risks are complex, difficult and diverse, mentioned in the introduction. The only part in line 39, gives a part of interface risks which may include designers, owners, project team members, main contractors, subcontractors, host communities, licensing and regulatory bodies, vendors, maintenance contractors and material supplies related issues.
Table 7 collects responses on work culture related to interface risks (4 questions, without stating clearly what interface risks are)
(2) Pretending that there are existed interface risks, one needs to understand how to manage them to make projects succeed. Subsection 2.2 reviews interface risk management from earlier works, Table 8, Table 9 and Table 10 present KMO and Bartlett's test on sampling adequacy and sphericity (No interface risk management).
Table 11 provides the research objective 1, Table 12 provides the research objective 2, and Table 13 provides the research objective 3. From survey sample responses, authors finish the research by counting responses to conclude research objectives. It is not clear what those numbers or percentages mean to research objectives and what they are contributed into the intrinsic structure of the complex interface risks.
(3) With the factor analysis and correlation matrices on 205 sample size, depicting the factor explanations on research objectives is limited and barely evident.
The manuscript describes a big plan for the research objectives, but lacks of enough information to identify and manage interface risks. Moreover, authors uses survey responses to basic counting answers to support the conclusion without any professional knowledge on the construction industry to link those factors and understanding the deep risks for completing projects. I find that the results of the manuscript are lack of reasoning and evident construction projects understanding, the method and research objectives are too elementary to support the conclusion, and overall the contribution of the manuscript is very limited. I would not be able to recommend this manuscript for publication on the journal CivilEng.

Author Response
Reviewer two
Comments and Suggestions for Authors
The manuscript is to provide a systematic approach on identifying and managing interface risks between project stakeholders in construction projects. The construction industry faces interface risks which are the most encountered problem in the construction industry. THe research objective illustrates three major objectives in this manuscript without identifying the interface risks. We need to first know what interface risks are, before we try to understand how to manage interface risks and understand what causes of interface risks are. More comments and suggestions are given below.
(1) As the title of the manuscript, one needs to identify interface risks between project stakeholders in construction projects. The interface risks are complex, difficult and diverse, mentioned in the introduction. The only part in line 39, gives a part of interface risks which may include designers, owners, project team members, main contractors, subcontractors, host communities, licensing and regulatory bodies, vendors, maintenance contractors and material supplies related issues.
Table 7 collects responses on work culture related to interface risks (4 questions, without stating clearly what interface risks are)
My response
Interfaces are points of interaction between two or more aspects of a project which might be between clients, contractors, subcontractors and other project stakeholders. Poor interface management gives rise to interface risks. Interface risks and the failure to manage them effectively, is a common cause of problems in construction projects which can negatively affect project objectives and goals.
This was also added before table 7 - Interfaces are points of interaction between two or more aspects of a project which might be between clients, contractors, subcontractors and other project stakeholders while interface risks are risks generated because of poor interface management in construction projects.
(2) Pretending that there are existed interface risks, one needs to understand how to manage them to make projects succeed. Subsection 2.2 reviews interface risk management from earlier works, Table 8, Table 9 and Table 10 present KMO and Bartlett's test on sampling adequacy and sphericity (No interface risk management).
Table 11 provides the research objective 1, Table 12 provides the research objective 2, and Table 13 provides the research objective 3. From survey sample responses, authors finish the research by counting responses to conclude research objectives. It is not clear what those numbers or percentages mean to research objectives and what they are contributed into the intrinsic structure of the complex interface risks.
My comment
Table 8 below shows the KMO and Bartlett’s test for research objective 1 (consequences of poor and ineffective interface risks management approach)
Table 8. KMO and Bartlett’s test for research objective 1 for B2 (consequences of poor and ineffective interface risks management approach).
Table 9. KMO and Bartlett’s test for research objective 2 for B3 (interface risks management approaches by organisations).
Table 10 below represents KMO and Bartlett’s test for research objective 3 (causes of interface risks)
Table 10. KMO and Bartlett’s test for research objective 3 for B4 (causes of interface risks).
The item statistics for the research objectives were added on the appendices. They were excluded initially to reduce the pages.
The theoretical and empirical factors reliability were confirmed and the focus was on the theoretical factors and they were reported on the paper.
Please see appendix A in the manuscript
(3) With the factor analysis and correlation matrices on 205 sample size, depicting the factor explanations on research objectives is limited and barely evident.
The manuscript describes a big plan for the research objectives, but lacks of enough information to identify and manage interface risks. Moreover, authors uses survey responses to basic counting answers to support the conclusion without any professional knowledge on the construction industry to link those factors and understanding the deep risks for completing projects. I find that the results of the manuscript are lack of reasoning and evident construction projects understanding, the method and research objectives are too elementary to support the conclusion, and overall the contribution of the manuscript is very limited. I would not be able to recommend this manuscript for publication on the journal CivilEng.
My comment
The questionnaire was distributed all over South Africa to active construction industry professionals who played vital roles in construction projects and 205 responses were received which were enough sample size for the study. The responses were useful in realising the research objectives.
Thank you for your recommendations. They were helpful and I learnt a lot. The changes were added to the paper.
Please see the attachment

Reviewer 3 Report
Comments and Suggestions for Authors
The manuscript is interesting and deserves the attention of academics and practitioners, but it suffers from some weaknesses that make it not suitable for publication in its curent form. My remarks are reported below.
1) The concept of interfacse is quite intuitive, but it deserves to be defined from the beginning (it appears at the end of Section 1).
2) I suggest reformulating the research questions by following a logical cause-effect sequence, that is causes before effects.
3) No method on the literatue search is provided in the related section.
4) Some details on KMO measure, Bartlett's test, Cronbach’s Alpha test and Spearman’s Rho could be interesting for the reader.
5) Section 4 is too spread and contains too many tables that limit its readibility. I suggest reducing the said section and focusing it only on the core elements for providing useful takeways. Other data can be reported in Appendix.
6) Figures 4 and 6 seem overlapped. In risk analysis terminology, risk management (or control) refers to the risk mitigation while the previous risk analysis stage, i.e., the risk assessment, refers to risk identification and quantification. I suggest using this framework to make the figures more understandable from a risk analysis perspective.
7) The manuscipt contains several typesetting mistakes that must be fixed.
Author Response
Reviewer three
Comments and Suggestions for Authors
The manuscript is interesting and deserves the attention of academics and practitioners, but it suffers from some weaknesses that make it not suitable for publication in its curent form. My remarks are reported below.
- The concept of interfacse is quite intuitive, but it deserves to be defined from the beginning (it appears at the end of Section 1).
My comment
Interfaces are points of interaction between two or more aspects of a project which might be between clients, contractors, subcontractors and other project stakeholders. Poor interface management gives rise to interface risks. Interface risks and the failure to manage them effectively, is a common cause of problems in construction projects which can negatively affect project objectives and goals.
- I suggest reformulating the research questions by following a logical cause-effect sequence, that is causes before effects.
My comment
To support the objectives of the study, these three research questions were asked. Respondents were asked to identify:
- The causes of interface risks
- The consequences of poor and ineffective interface risks management approach
- The interface risks management approaches implemented by their organisations
3) No method on the literatue search is provided in the related section.
4) Some details on KMO measure, Bartlett's test, Cronbach’s Alpha test and Spearman’s Rho could be interesting for the reader.
My comment
To establish the consistency of data, the value of the Cronbach’s Alpha (coefficient alpha was determined). Table 26 below shows the reliability statistics for research objective 1, Cronbach’s alpha coefficients must be greater than 0,7 to confirm reliability and internal consistency.
KMO seeks to determine the applicability of a result to a set of measures when conducting Factor analysis and the values must be greater than 0.6 while Bartlett's Test of Sphericity must be less than 0.05 to establish the applicability of factor analysis.
5) Section 4 is too spread and contains too many tables that limit its readibility. I suggest reducing the said section and focusing it only on the core elements for providing useful takeways. Other data can be reported in Appendix.
6) Figures 4 and 6 seem overlapped. In risk analysis terminology, risk management (or control) refers to the risk mitigation while the previous risk analysis stage, i.e., the risk assessment, refers to risk identification and quantification. I suggest using this framework to make the figures more understandable from a risk analysis perspective.
My comment
Figure 6 is the detailed interface risk minimisation framework that incorporates stakeholder management.
Figure 4 is the proposed interface risk assessment approach framework.
The bigger tables were moved to the appendices
7) The manuscript contains several typesetting mistakes that must be fixed.
Typesetting errors fixed
Thank you for your suggestions and recommendations. They were helpful and I learnt a lot. The changes were added to the paper.
Please see the attachment

Round 2
Reviewer 1 Report
Comments and Suggestions for Authors
Thank you for addressing the comments.
Comments on the Quality of English LanguageEnglish is fine overall.
Reviewer 2 Report
Comments and Suggestions for Authors
Authors response my previous concerns carefully and clearly.
Under the clarification of the interface risk and its management, authors collect 205 responses from all over South Africa to carry this study.
1. Line 529, Cronbach's alpha should be better to add valid cases, Cronbach's alpha based on Standardrized items, and SPSS operation explanations;
2. Figure 4, identifying/recommending interface risks has any intrinsic relation or logic behind the simple tree expanded out?
3. Figure 5 will be better to indicate the systematic approach for proposed stakeholder management with number labels or arrow direction to show the importance;
4. Figure 6 should be given a tree-like mitigation method on interface risks, provided with each part to mitigate specific interface risk on each box.
Reviewer 3 Report
Comments and Suggestions for Authors
All my comments have been addressed properly.
Best regards